# The transaminase-ω-amidase pathway senses oxidative stress to control glutamine metabolism and α-ketoglutarate levels in endothelial cells

Niklas Herrle[1,2,28], Pedro F Malacarne [1,2,28], Timothy Warwick [1,2], Alfredo Cabrera-Orefice[1,3], Yiheng Chen[4,5], Maedeh Gheisari[1,2], Souradeep Chatterjee [1,2], Matthias S Leisegang [1,2], Tamim Sarakpi[6,7], Sarah Wionski[1,2], Melina Lopez [1,2], Carine Kader [1,2], Tom Teichmann[1,2], Maria-Kyriaki Drekolia[8], Ina Koch [9], Marcus Keßler [9], Sabine Klein[10], Frank Erhard Uschner[10], Jonel Trebicka[10], Steffen Brunst[11], Ewgenij Proschak[11], Stefan Günther [12], Mónica Rosas-Lemus[13], Nina Baumgarten [2,14], Stephan Klatt [2,15], Thimoteus Speer[6,7], Sofia-Iris Bibli[8], Marta Segarra [16], Amparo Acker-Palmer [16], Julian U G Wagner[2,17], Ilka Wittig[1,3], Stefanie Dimmeler[2,17], Marcel H Schulz [2,14], J B Richards[4,5,18,19,20,21], Ralf Gilsbach[22], Travis T Denton[23,24,25], Ingrid Fleming [2,15], Luciana Hannibal[26,27], Ralf P Brandes [1,29✉] & Flávia Rezende [1,2,29✉]

## Abstract

Oxidative stress is a major driver of cardiovascular disease; however, the fast changes in cellular metabolism caused by short-lived reactive oxygen species (ROS) remain ill-defined. Here, we characterized changes in the endothelial cell metabolome in response to acute oxidative challenges and identified novel redox-sensitive metabolic enzymes. $H_2O_2$ selectively increased the amount of α-ketoglutaramate (αKGM), a largely uncharacterized metabolite produced by glutamine transamination and an unrecognized intermediate of endothelial glutamine catabolism. In addition, $H_2O_2$ impaired the catalytic activity of nitrilase-like 2 ω-amidase (NIT2), the enzyme that converts αKGM to α-ketoglutarate (αKG), by the reversible oxidation of specific cysteine residues. Moreover, a *NIT2* gene variant exhibited decreased expression in humans and was associated with increased plasma αKGM concentration. Endothelial-specific knockout of NIT2 in mice increased cellular αKGM levels and impaired angiogenesis. Further, NIT2 depletion impaired endothelial cell proliferation, sprouting, and induced senescence. In conclusion, we uncover NIT2 as a redox-sensitive enzyme of the glutamine transaminase-ω-amidase pathway that acts as a metabolic switch modulating endothelial glutamine metabolism in mice and humans.

Keywords Glutamine Metabolism; Oxidative Stress; Endothelial Cells; α-Ketoglutarate; α-Ketoglutaramate
Subject Categories Cardiovascular System; Metabolism; Vascular Biology & Angiogenesis

[1]Goethe University, Institute for Cardiovascular Physiology, Frankfurt am Main, Germany. [2]German Center of Cardiovascular Research (DZHK), Partner Site Rhein Main, Frankfurt am Main, Germany. [3]Goethe University, Functional Proteomics, Frankfurt am Main, Germany. [4]McGill University, Department of Human Genetics, Montréal, QC, Canada. [5]McGill University, Lady Davis Institute, Jewish General Hospital, Montréal, QC, Canada. [6]Goethe University, Department of Internal Medicine 4, Nephrology, Frankfurt am Main, Germany. [7]Else Kroener-Fresenius Center for Nephrological Research, Frankfurt am Main, Germany. [8]Heidelberg University, Department of Vascular Dysfunction, Mannheim, Germany. [9]Goethe University, Institute of Computer Science, Frankfurt am Main, Germany. [10]University Hospital Münster, Department of Internal Medicine B, Münster, Germany. [11]Goethe University, Institute for Pharmaceutical Chemistry, Frankfurt am Main, Germany. [12]Max-Planck-Institute for Heart and Lung Research, Bad Nauheim, Germany. [13]University of New Mexico, Department of Molecular Genetics and Microbiology, Health Sciences Center, Albuquerque, NM, USA. [14]Goethe University, Institute for Computational Genomic Medicine, Frankfurt am Main, Germany. [15]Goethe University, Institute for Vascular Signaling, Frankfurt am Main, Germany. [16]Goethe University, Institute of Cell Biology and Neuroscience, Frankfurt am Main, Germany. [17]Goethe University Frankfurt, Institute for Cardiovascular Regeneration, Frankfurt am Main, Germany. [18]McGill University, Department of Epidemiology, Biostatistics and Occupational Health, Montréal, QC, Canada. [19]5 Prime Sciences Inc, Montréal, QC, Canada. [20]McGill University, Department of Medicine, Montréal, QC, Canada. [21]King's College London, Department of Twin Research, London, UK. [22]Heidelberg University Hospital, Institute of Experimental Cardiology, Heidelberg, Germany. [23]Washington State University Health Sciences Spokane, Department of Pharmaceutical Sciences, College of Pharmacy and Pharmaceutical Sciences, Spokane, WA, USA. [24]Washington State University Health Sciences Spokane, Department of Translational Medicine and Physiology, Elson S. Floyd College of Medicine, Spokane, WA, USA. [25]Washington State University Health Sciences Spokane, Steve Gleason Institute for Neuroscience, Spokane, WA, USA. [26]University of Freiburg, Laboratory of Clinical Biochemistry and Metabolism, Department of General Pediatrics, Adolescent Medicine and Neonatology, Freiburg im Breisgau, Germany. [27]University of Freiburg, CIBSS – Centre for Integrative Biological Signalling Studies, Freiburg, Germany. [28]These authors contributed equally as first authors: Niklas Herrle, Pedro F Malacarne. [29]These authors contributed equally as senior authors: Ralf P Brandes, Flávia Rezende.✉E-mail: brandes@vrc.uni-frankfurt.de; rezende@vrc.uni-frankfurt.de

# Introduction

Oxidative stress is a hallmark and a potential driver of cardiovascular diseases. Reactive oxygen species (ROS) are a heterogeneous class of molecules, which differ in their reactivity, biological targets, and functional relevance. Nitric oxide, superoxide anions ($O_2^{\bullet-}$), and hydrogen peroxide ($H_2O_2$) are of particular importance due to their relevance in cellular signaling (Sies et al, 2022). Cellular responses to an acute challenge with ROS have been mainly studied regarding signal transduction and gene expression. Because ROS are short-lived molecules, fast changes in the cellular metabolome are at the forefront of the response to oxidative stress (Ralser et al, 2009). The characterization of redox regulation of metabolism remains incomplete, despite significant advances in the field of metabolomics. Well established examples of redox-control of metabolism are largely restricted to the best studied metabolic pathways such as the superoxide-sensitive iron–sulfur clusters of aconitase and isocitrate dehydrogenase (ACO1, IDH1, tricarboxylic acid (TCA) cycle) (Gardner, 2002a) as well as glucose-6-phosphatase dehydrogenase (G6PD, pentose phosphate pathway) (Kuehne et al, 2015), glyceraldehyde 3-phosphate dehydrogenase (GAPDH, glucose oxidation)(Jeong et al, 2011) and pyruvate kinase M2 (PKM2, glycolysis) (Anastasiou et al, 2011).

The metabolic plasticity of endothelial cells is unique; as depending on their environment, they can be proliferative, quiescent, stationary, or migratory and, thus, are exposed to high or low partial pressure of oxygen. Endothelial cells are also a prime site of inflammation and, as such, are exposed to changes in their redox environment or even overt oxidative stress. Although an oversimplification, it has been reported that endothelial cells re-generate ATP mainly via glycolysis while oxidizing only <1% of the pyruvate generated in TCA cycle (de Bock et al, 2013; Pasut et al, 2021; Dumas et al, 2020). In human umbilical vein endothelial cells (HUVEC), the glycolytic flux is more than 200-fold higher than glucose oxidation in the electron transport chain (Eelen et al, 2015; Eelen et al, 2018; Wong et al, 2017). This is supported by the fact that endothelial cells have few mitochondria and generate nitric oxide, which inhibits mitochondrial respiration. This inhibition reduces oxidative phosphorylation, oxygen consumption, and mitochondrial ROS generation (Bailey et al, 2019). Thus, endothelial cells utilize glutamine as an anaplerotic source of carbon for the biosynthesis of nucleic acids, lipids, and other building blocks required for proliferation (Kim et al, 2017; Huang et al, 2017). Within the cell, glutamine is converted to glutamate by glutaminase (GLS1) and thereafter to α-ketoglutarate (αKG) by glutamate dehydrogenase (GLUD1) or by transamination with a suitable α-keto acid substrate. This reaction is known as the glutaminase I pathway, and αKG is a central TCA cycle intermediate and at the crossroads of several metabolic processes. Based on the metabolic features of endothelial cells, we analyzed the endothelial metabolome in response to an acute oxidative challenge with menadione (to generate $O_2^{\bullet-}$) or extracellular $H_2O_2$. We observed that $H_2O_2$ selectively unmasked a redox-sensitive and functionally important non-canonical pathway for the generation of αKG from glutamine.

# Results

## $H_2O_2$ increases α-ketoglutaramate levels in endothelial cells

To assess the metabolic response of endothelial cells to menadione and $H_2O_2$, we utilized our previously published dataset, which compared the time-resolved responses to various types of ROS (Müller et al, 2022). Menadione, a redox cycler that generates intracellular $O_2^{\bullet-}$, led to a significant decrease in isocitrate levels consistent with the known inhibition of aconitase by superoxide anions (Gardner, 2002b). However, the levels of downstream metabolites of αKG were not affected. This can be explained by the fact that glutamine, through GLS1, replenishes carbons via αKG into the TCA cycle (Fig. 1A,C). Strikingly, $H_2O_2$ selectively decreased the levels of αKG and its downstream metabolites in the TCA cycle but also conversely increased the levels of α-ketoglutaramate (αKGM, Fig. 1B–E). This largely uncharacterized metabolite is rarely mentioned in the literature because it is not commercially available and, thus, not detected in targeted mass spectrometry-based analysis (LC-MS/MS) (Shurubor et al, 2016). αKGM is formed by transamination of glutamine by the transaminase enzymes KYAT1 and KYAT3 (kynurenine aminotransferases, previously annotated in the human genome as GTK and GTL for glutamine transaminase of kidney and liver, respectively) (Pinto et al, 2014). αKGM can be converted to αKG through enzymatic de-amidation. If not metabolized, αKGM rapidly degrades into a stable inert lactam: 2-hydroxy-5-oxo-proline. Given its low stability, αKGM therefore does not occur in relevant concentrations in biological systems, and rather the lactam product accumulates and is detected as a footprint of αKGM. Since $H_2O_2$ increased αKGM and decreased αKG in endothelial cells in the present study, this reaction likely involves an enzyme that is redox-sensitive and inhibited by $H_2O_2$.

## De-amidation of αKGM to αKG is catalyzed by NIT2

The de-amidation of αKGM to αKG is catalyzed by the nitrilase-like 2, ω-amidase enzyme, NIT2. The non-canonical reaction cycle to generate αKG from glutamine is referred to as the glutamine transaminase-ω-amidase pathway (GTωA) (Fig. 1F) (Denton and Cooper, 2023). This pathway was proposed in the 1950s (Meister, 1953; Meister and Otani, 1957) but has been only partially characterized and since then unrecognized. Importantly, no knockout cell nor mouse model of the pathway has been studied up to now, and its physiological relevance is largely unclear. We therefore decided to further focus on the metabolite αKGM and the biological relevance of the pathway.

αKGM is contained in an untargeted metabolite panel of Metabolon® (as determined by its fragmentation pattern in the lactam form) but LC-MS/MS confirmation on the basis of standards has not been performed. Chemical synthesis of αKGM yields, as expected, only the lactam form (Shen et al, 2020), which is here referred to αKGM. LC-MS/MS validation (level 1; according to metabolomics standard initiative) showed three major fragmentation peaks (negative mode) at 126.0 Da, 82.0 Da and 42.0 Da (Fig. EV1A). αKGM was also detected in positive mode yielding a fragments of 105.1 and 91 Da (data not shown). Using pure αKGM

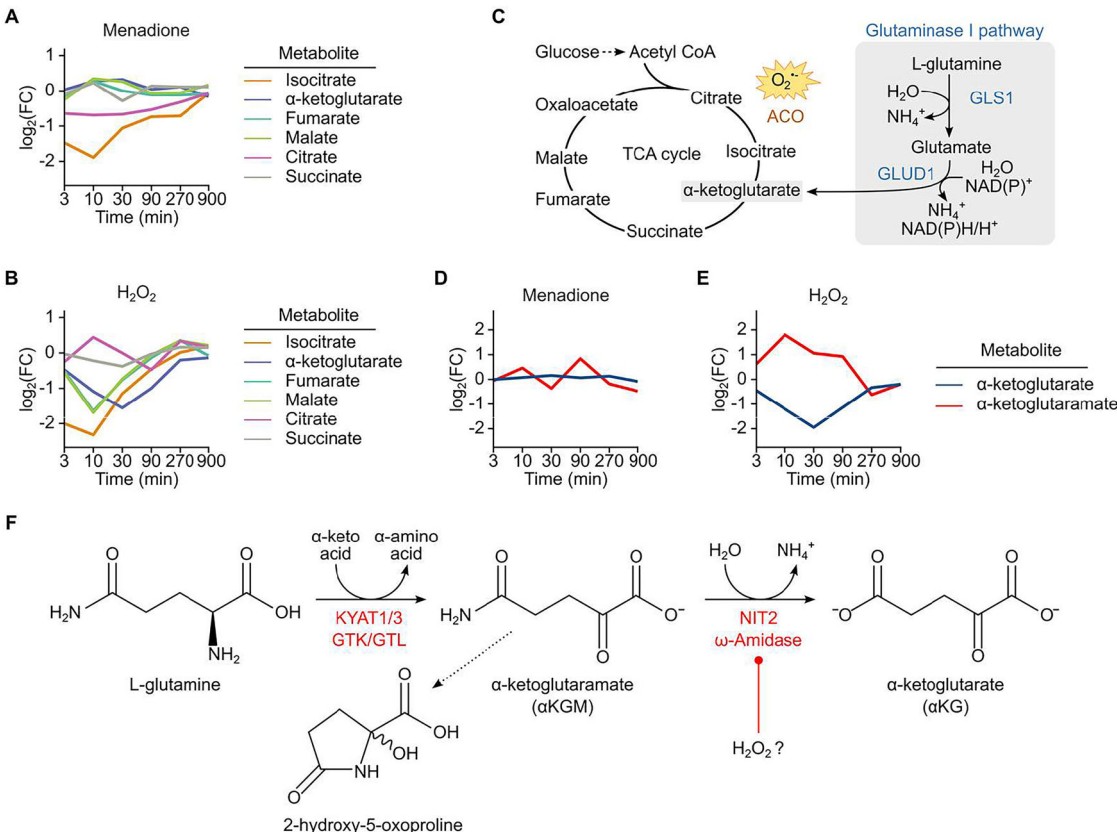

**Figure 1. Exposure of endothelial cells to H₂O₂ increases α-ketoglutaramate (αKGM).**

Mean of $\log_2$ fold change in TCA cycle metabolites (untargeted metabolomics) of human umbilical endothelial cells (HUVEC) exposed to either menadione (5 µM) (**A**) or H₂O₂ (300 µM) (**B**). (**C**) Superoxide oxidizes the iron–sulfur cluster in aconitase; however, the glutaminase I pathway can replenish α-ketoglutarate (αKG) into the TCA cycle through the action of glutaminase 1 (GLS1). (**D, E**) Changes in αKGM and αKG in HUVEC after exposure to menadione or H₂O₂. (**F**) Reactions of the glutamine transaminase-ω-amidase (GTωA) pathway. ACO aconitase, GLUD1 glutamate dehydrogenase 1, KYAT1 kynurenine aminotransferase 1 or glutamine transaminase of kidney, KYAT3 kynurenine aminotransferase 3 or glutamine transaminase of liver, NIT2 nitrilase-like 2, ω-amidase. Source data are available online for this figure.

as a standard, we determined its concentration in plasma and urine from healthy individuals. The average plasma levels of αKGM and αKG were 3.4 and 12.3 µM, respectively. In urine, the concentrations were 19.6 and 9.3 µmol/mmol of creatinine for αKGM and αKG, respectively (Fig. EV1B).

## A single-nucleotide variant (SNV) decreases NIT2 expression and elevates plasma levels of αKGM in humans

To determine a potential correlation between NIT2 and αKGM levels in humans, we performed a metabolite quantitative trait locus analysis (mQTL) along with an expression quantitative trait locus analysis (eQTL). For this analysis, we used datasets (Chen et al, 2023) generated by Metabolon® and we confirmed the identity of αKGM using LC-MS/MS. One sentinel SNV (rs3830303, chr3: 100,334,840:C/GC) showed a highly significant association with increased plasma αKGM ($P$ value $1.4 \times 10^{-36}$) (Chen et al, 2023). However, this SNV localizes in an indel and is not present in many genome-wide association studies. The second most significant SNV is rs277627 (chr3: 100,336,429: G/A) that is in linkage distribution to rs3830303 and has a $P$ value of $2.0 \times 10^{-36}$ for increased αKGM

(Fig. 2A). rs277627 is located in intron 1 of the *NIT2* gene, which contains a regulatory element (REM, enhancer coordinates: chr3: 100,336,401–100,336,500, hg38) that can potentially affect the binding of transcription factors (TF). Of the 15 transcription factors that bind to this regulatory element (according to SNEEP (Baumgarten et al, 2024)) and are expressed in human endothelial cells (RNAseq data, Dataset 1), we observed that the binding of nine of them was likely to be affected by the mutation (Fig. EV2A). To verify if rs277627 affects NIT2 expression, we deleted the REM using CRISPR/cas9 in HEK 293 cells (Figs. 2B,C and EV2B). Deletion of the locus containing rs277627 resulted in a decrease in expression of NIT2 mRNA and protein (Fig. 2D,E). In line with this, the rs277627 GA and AA variants resulted in a lower expression of NIT2 in human aorta and tibial artery samples (GTEx data(GTEx Consortium 2020), Fig. EV2C), than subjects carrying GG. Likewise, CRISPR/Cas9-mediated homology-directed repair to generate the NIT2 intron containing the rs277627 (G → A) in HUVEC led to a reduction of NIT2 RNA expression (Figs. 2F and EV2D,E).

Phenome-wide association studies to rs3830303 and rs277627 (Datasets 2 and 3) revealed a positive association of rs277627 with risk factors for hypertension (odds ratio: 1.06, $P = 0.01$; CLSA

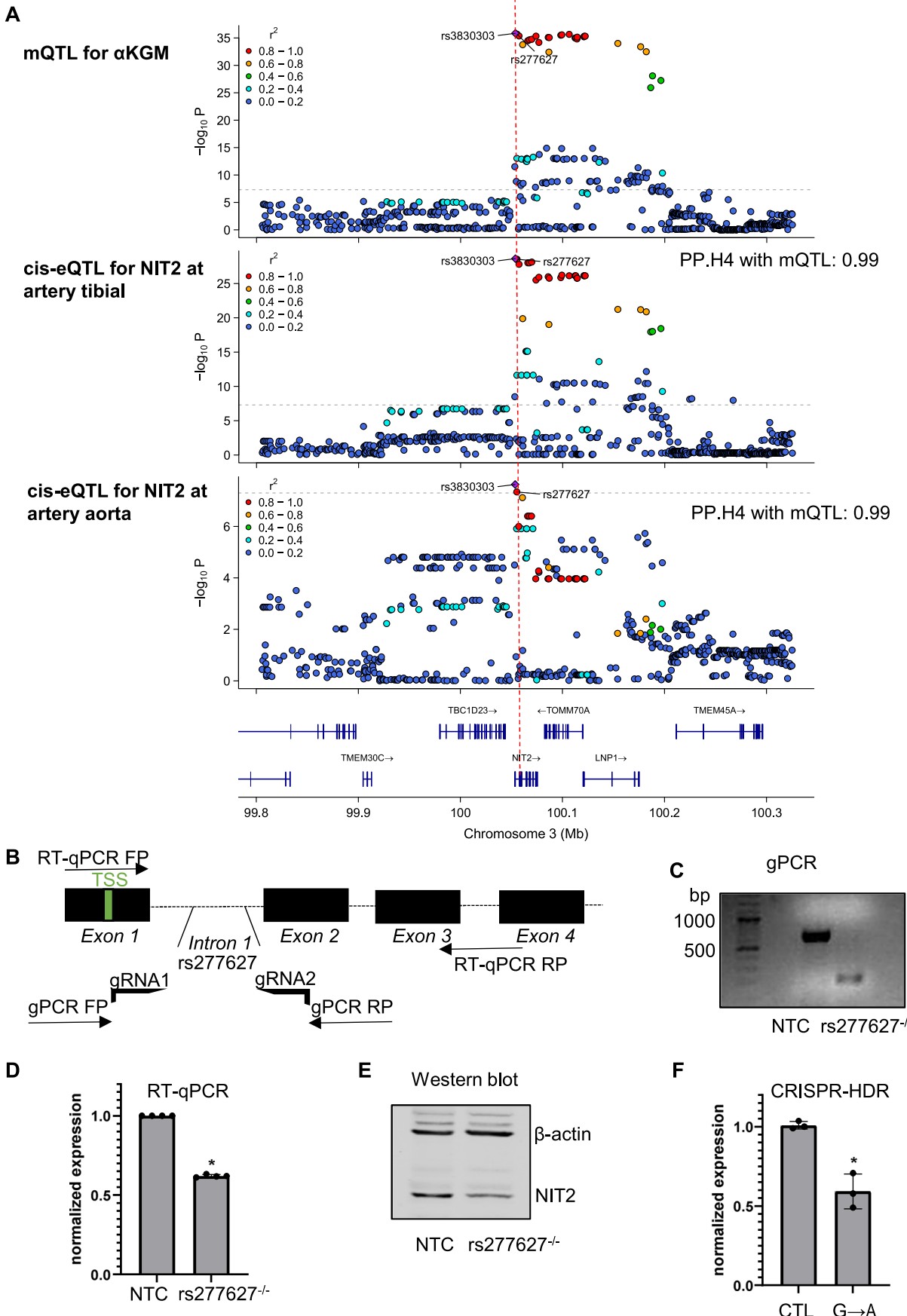

**A**

mQTL for αKGM

cis-eQTL for NIT2 at artery tibial

PP.H4 with mQTL: 0.99

cis-eQTL for NIT2 at artery aorta

PP.H4 with mQTL: 0.99

TBC1D23→    ←TOMM70A    TMEM45A→

TMEM30C→    NIT2→    LNP1→

Chromosome 3 (Mb)

**B**
RT-qPCR FP
TSS
Exon 1    Intron 1    Exon 2    Exon 3    Exon 4
rs277627    RT-qPCR RP
gRNA1    gRNA2
gPCR FP    gPCR RP

**C** gPCR
bp
1000
500
NTC    rs277627⁻ᐟ⁻

**D** RT-qPCR
normalized expression
NTC    rs277627⁻ᐟ⁻

**E** Western blot
β-actin
NIT2
NTC    rs277627⁻ᐟ⁻

**F** CRISPR-HDR
normalized expression
CTL    G→A

**Figure 2. A human single-nucleotide variant (SNV) decreases *NIT2* expression and elevates plasma levels of αKGM.**

(A) Co-localization analyses and regional association plots of αKGM mQTL (CLSA metabolite, $n = 8203$) with cis-eQTL for *NIT2* in artery tibial ($n = 475$) and aortic tissue ($n = 329$) (GTEx v8 study). The sentinel variants rs3830303 and rs277627 are indicated. (B) Strategy for CRISPR/cas9 deletion of the regulatory element containing rs277627 in HEK 293 cells. (C) Genomic PCR showing a 437 bp deletion of the rs277627 containing locus in one single clone (after clonal expansion of HEK 293). *NIT2* expression in NTC and rs27767$^{-/-}$ HEK 293 cells by RT-qPCR. $P = 0.0286$, Mann–Whitney test. (D) and Western blot (E). (F) RT-qPCR of NIT2 after CRISPR/Cas9-mediated homology-directed repair to generate the NIT2 intron containing SNP rs277627 (G → A). HUVEC CTL were electroporated with Cas9-GFP only. *$P = 0.003$ paired *t* test. Expression is normalized to beta-actin. TSS transcription start site. RT-qPCR real-time polymerase chain reaction, FP forward primer, RP reverse primer, NTC non-targeting control. Source data are available online for this figure.

cohort, analyzed with logistic regression. Fasting hour, sex, age, BMI, and recruitment centers adjusted in the model) (Chen et al, 2023). Thus, a variant of the *NIT2* gene that leads to the accumulation of αKGM in the plasma, resulted in decreased NIT2 mRNA and protein levels in humans.

## NIT2 and GLS1 synergistically maintain the endothelial metabolome

Given the lack of knowledge on the relevance of the pathway in cells, we set out to investigate its contribution to glutamine metabolism in conjunction with the endothelial default pathway involving GLS1. CRISPR/cas9 was used to generate endothelial cells lacking NIT2 (NIT2$^{-/-}$) and GLS1 (GLS1$^{-/-}$) alone or in combination (NIT2/GLS1$^{-/-}$). Western blot analysis and immuno-fluorescence staining confirmed successful knockout (Fig. 3A,B). Furthermore, immunofluorescence suggested that GLS1 has a mitochondrial localization as previously described (Kim et al, 2017), whereas NIT2 is distributed across the cell (Fig. 3B).

To monitor the levels of αKGM and αKG, targeted LC-MS/MS measurements were performed. αKGM was not detected in either endothelial cells treated with a non-targeted construct (NTC) or in cells lacking GLS1, presumably as a result of high NIT2 activity. However, the deletion of NIT2 resulted in a marked increase in αKGM levels (Fig. 3C), accompanied by a slight decrease in αKG, which was more markedly affected by the deletion of GLS1 (Fig. 3D). To analyze glutamine utilization by the two pathways, we performed isotopic tracing with fully labeled glutamine ($^{13}C_5$, $^{15}N_2$) and utilized a heavy isotope of αKGM (m + 5, m + 1) as reference (Fig. 3E). The knockout of NIT2 increased the abundance of m + 5, m + 1 αKGM by 4 fold (Fig. 3F) but did not affect the abundance of m + 5 αKG. In contrast, deletion of GLS1 had no impact on m + 5, m + 1 αKGM levels but largely contributed to m + 5 αKG (Fig. 3G). Importantly, the double knockout of NIT2 and GLS1 further decreased m + 5 αKG as compared to the single GLS1 knockout. These observations suggest that despite the large contribution of GLS1 to αKG levels in endothelial cells, NIT2 functionally generates αKG from glutamine. NIT2-derived αKG production seems to be particularly relevant when GLS1 activity is low or under acute inhibition of NIT2.

To explore the overall contribution of NIT2 and GLS1 to the endothelial cell metabolome, we performed untargeted metabolomics of single and double knockout cells. αKGM stood out as the most significantly increased metabolite of those measured in NIT2$^{-/-}$ and NIT2/GLS1$^{-/-}$ cells (Fig. 3H). The deletion of NIT2 significantly altered 10 unique metabolites, whereas the knockout of GLS1 altered 51 metabolites. However, the combined deletion of both enzymes significantly altered 86 unique metabolites, suggesting that both pathways synergistically contribute to the

metabolome of endothelial cells (Dataset 4). A similar trend was observed regarding gene expression as the deletion of NIT2 differentially regulated seven unique genes, whereas GLS1$^{-/-}$ altered 91 and NIT2/GLS1$^{-/-}$ 1161 (Fig. EV3; Dataset 1).

Altogether, deletion of NIT2 led to an accumulation of αKGM and a decrease in αKG levels. GLS1 has a greater contribution to αKG production than NIT2, but both pathways are active and maintain the metabolome and gene signature of endothelial cells.

## Endothelial knockout of NIT2 impairs angiogenesis in mice

To explore the function of NIT2 in vivo, we generated endothelial cell-specific, tamoxifen-inducible knockout mice of NIT2 (CTL and ecNit2$^{-/-}$, Fig. EV4A,B). Efficient knockout of NIT2 in ecNit2$^{-/-}$ was confirmed by Western blot of endothelial cells enriched from the aorta (Fig. 4A). To evaluate the contribution of endothelial NIT2 to the glutamine system in mice, untargeted metabolomics was performed from plasma and lung tissue, which is rich in endothelial cells. In plasma, there was a trend toward increased αKGM ($P = 0.0506$) in ecNit2$^{-/-}$ as compared to control mice. Importantly, αKG was significantly decreased in ecNit2$^{-/-}$ as compared to CTL mice, demonstrating a contribution of endothelial NIT2 to plasma αKG level (Fig. 4B,C; Dataset 5). Untargeted metabolomics from whole lung showed αKGM as the second most upregulated metabolite in ecNit2$^{-/-}$ as compared to CTL mice and although αKG was unchanged, there was a significant increase in glutamine level, suggesting either substrate accumulation in the pathway or compensation (Fig. 4D; Dataset 6). Altogether, the metabolomics of ecNIT2$^{-/-}$ mice suggest a contribution of the endothelium to cellular and systemic αKGM and αKG. Moreover, the knockout of NIT2 in mice affects metabolites outside of the GTωA pathway.

For endothelial GLS1, an important contribution to angiogenesis is established (Huang et al, 2017). Thus, we examined whether deletion of NIT2 leads to this phenotype using the neonatal retina model of angiogenesis. Endothelial-specific, inducible knockout of NIT2 led to a decrease in vessel area and total vessel length. It also decreased endothelial cell proliferation (as demonstrated by EdU, 5-ethynyl-2'-deoxyuridine, incorporation) and decreased the number of vessel junctions as compared to retinae of CTL mice (Fig. 4E–K). Furthermore, in the aortic outgrowth assay (ex vivo model of angiogenesis), VEGF-induced sprouting was significantly decreased in ecNit2$^{-/-}$ as compared to CTL mice (Fig. EV4C,D).

A global, constitutive knockout mouse of NIT2 was also generated (Nit2$^{ko/ko}$, which does not express Cre recombinase, Fig. EV4E). Importantly, these mice exhibited decreased retinal angiogenesis, excluding any toxic effects of Cre recombinase activity (Horvath et al, 2024) (Fig. EV4F,G). To address the role

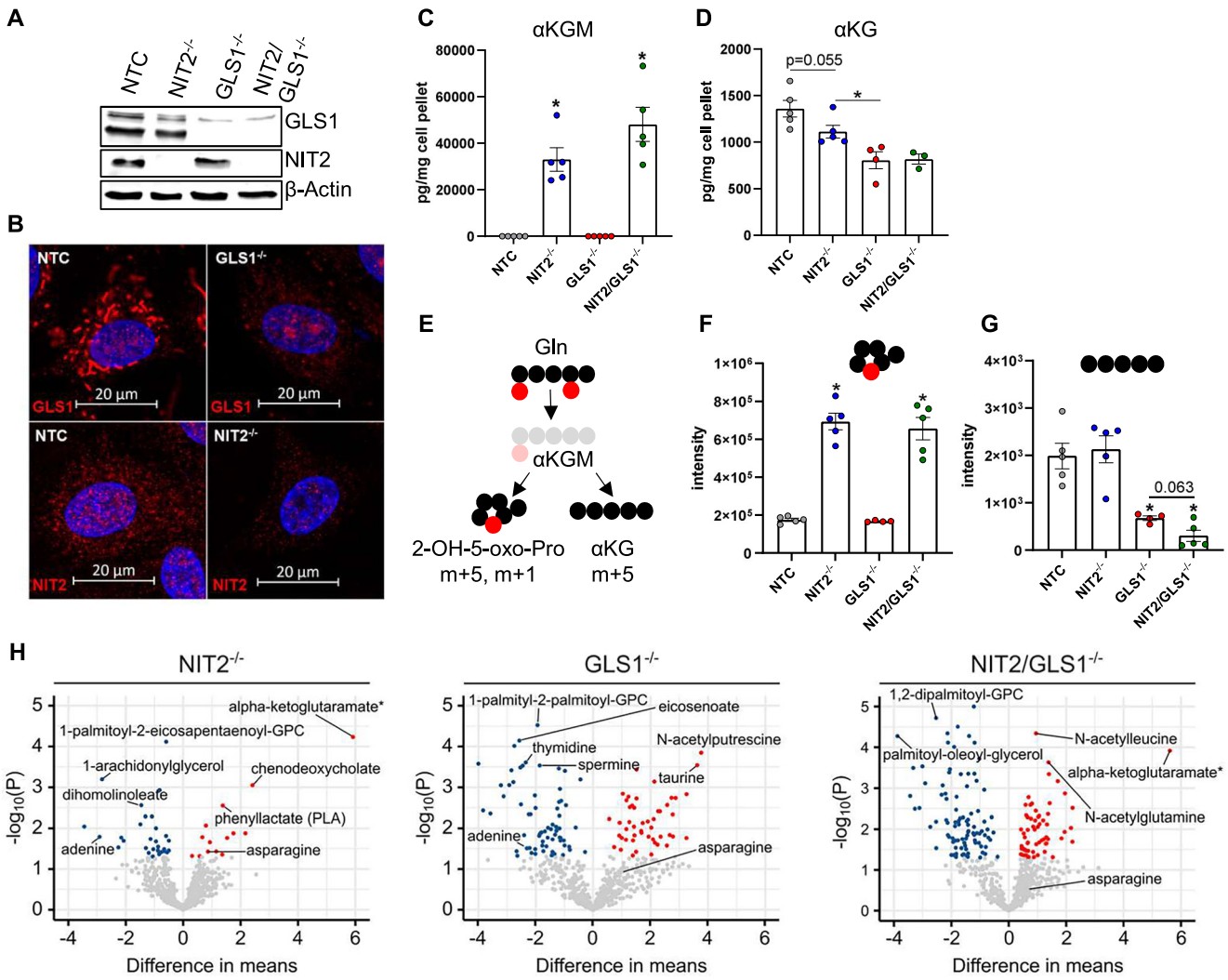

**Figure 3. NIT2 and GLS1 synergistically maintain the endothelial metabolome.**

NIT2$^{-/-}$, GLS1$^{-/-}$, and NIT2/GLS1$^{-/-}$ HUVEC were generated by CRISPR/cas9 and knockout efficiency was validated by Western blot (A). Cellular distribution of NIT2 and GLS1 as shown by immunofluorescence (B). (C, D) Targeted LC-MS/MS measurements for αKGM and αKG in CRISPR/cas9 HUVEC. *$P < 0.05$ as compared to NTC, ANOVA with Bonferroni correction. (E) Scheme for isotopic tracing of fully labeled glutamine that generates m + 5, m + 1 αKGM and m + 5 αKG. (F) m + 5, m + 1 αKGM (*$P < 0.05$ as compared to NTC, one-way ANOVA with Bonferroni correction) and m + 5 αKG (G) in HUVEC (P value as indicated for Mann–Whitney test comparing GLS1$^{-/-}$ versus NIT2/GLS1$^{-/-}$ cells. (H) Volcano plots of significantly altered metabolites as measured by untargeted metabolomics. NTC non-targeting control. Source data are available online for this figure.

of endothelial NIT2 for angiogenesis in vivo in the adult stage, we analyzed the choroidal neovascularization upon a laser-induced injury. Endothelial cells in this type of angiogenesis penetrate through Bruch's membrane into the normally avascular subretinal space (Gong et al, 2015). Additionally, in this model, vascularization was decreased upon endothelial deletion of NIT2 (Fig. 4L,M). In conclusion, endothelial knockout of NIT2 decreases angiogenesis, likewise endothelial deletion of GLS1.

## Deletion of NIT2 depletes adenine and induces senescence in cultured endothelial cells

Given the fact that αKGM in the lactam form is a stable compound, it might be a metabolic end-product. To investigate this, we

incubated HUVEC with αKGM (300 μM) for 4 or 24 h and performed an RNAseq. αKGM did not significantly alter gene expression as compared to untreated cells (Fig. EV5), suggesting that αKGM has no signaling function and is secreted from cells and excreted from the urine. Therefore, it is highly unlikely that the attenuated angiogenesis after deletion of NIT2 is a consequence of the accumulation of lactam αKGM. It is, therefore, plausible to assume that loss of NIT2 activity results in a shortage of other essential metabolites. To study this aspect, functional experiments were performed in HUVEC. First, we recapitulated the angiogenic role of NIT2 using the spheroid outgrowth assay. NIT2$^{-/-}$, similarly to GLS1$^{-/-}$ and NIT2/GLS1$^{-/-}$, impaired endothelial cell sprouting under basal but also the VEGF-A-induced sprouting (Fig. 5A–C). Given that angiogenic function in this assay is a consequence of

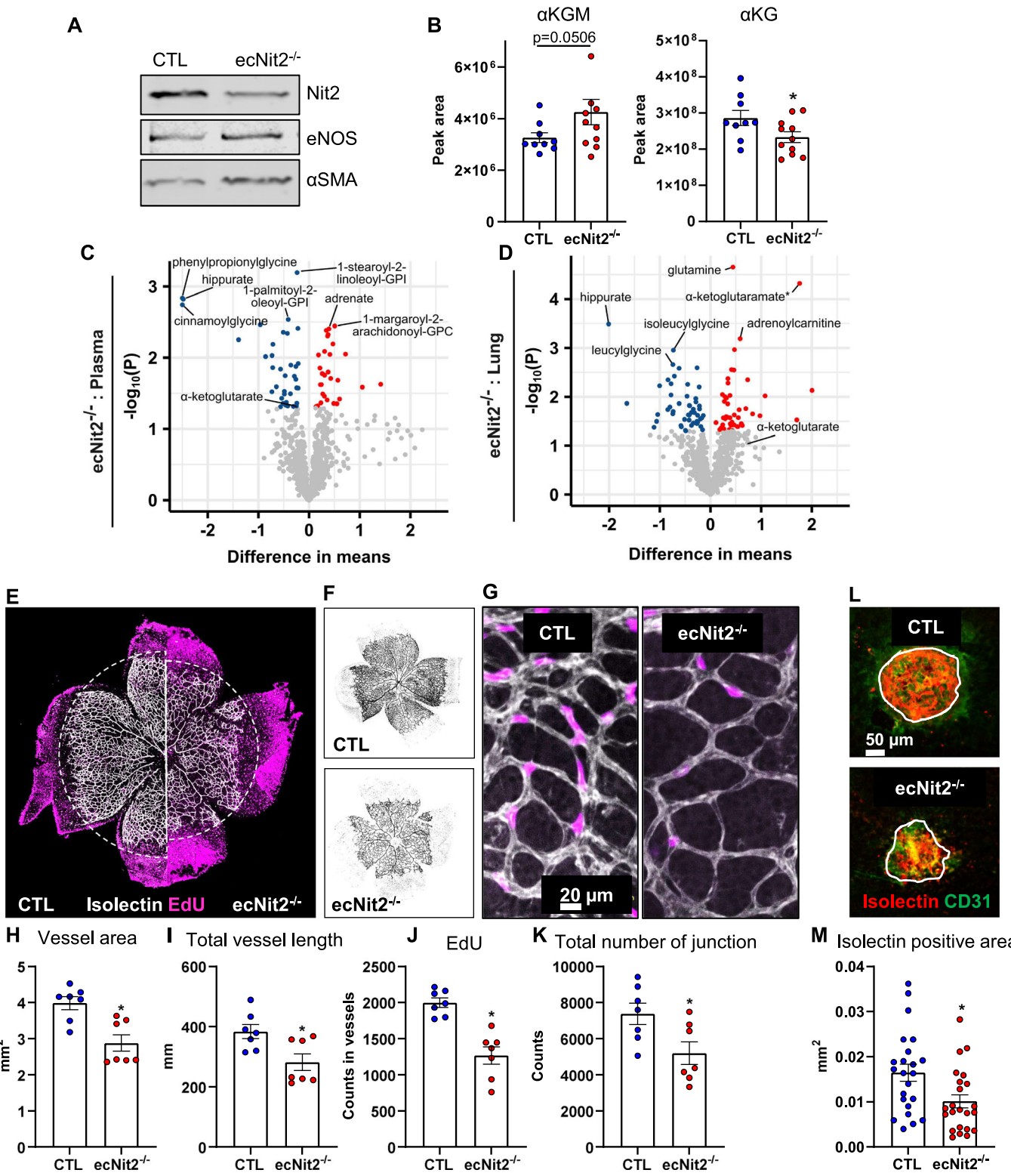

proliferative as well as migratory capacity of the endothelial cells, these aspects were differentiated. Neither knockout of NIT2 nor pharmacologic inhibition of GLS1 affected endothelial cell migration in the scratch wound assay (Fig. 5D). In contrast to this, both interventions decreased proliferation (Fig. 5E) with the

combination of NIT2 knockout and GLS1 inhibition having an additive effect. Impaired proliferation with maintained migration may imply that the synthetic function of the cells is attenuated. This could be a consequence of a lack of anaplerotic TCA cycle equivalents. However, it has been previously reported that

◀ **Figure 4. Endothelial knockout of NIT2 impairs angiogenesis in mice.**

Generation of a tamoxifen-inducible, endothelial cell-specific knockout mouse of Nit2 (ecNit2$^{-/-}$). (A) Validation of the knockout efficiency by Western blotting of aortic endothelial cells. (B) LC-MS/MS for αKGM and αKG in plasma of CTL and ecNit2$^{-/-}$ mice. *$P < 0.05$ as compared to CTL, Mann–Whitney test. (C, D) Untargeted metabolomics from CTL and ecNit2$^{-/-}$ mice from lung tissue (C) and plasma (D). (E) Retina angiogenesis in neonatal mice (P6), isolectin and EdU (5-ethynyl-2′-deoxyuridine) staining from CTL and ecNit2$^{-/-}$ mice. (F) Representative images of retinas from CTL and ecNit2$^{-/-}$ mice stained with Isolectin. (G) In all, ×40 magnification images of vessels between retinal arteries and veins. (H–K) quantification of vascular parameters as indicated. *$P < 0.05$, Mann–Whitney test. (L, M) Choroidal neovascularization upon a laser-induced injury in adult mice. Quantification of isolectin staining. *$P < 0.05$, Mann–Whitney test. Source data are available online for this figure.

decreased proliferation in glutamine-deprived endothelial cells was only partially rescued by adding extra TCA carbons in the form of cell-permeable compounds (dimethyl-αKG, monomethyl-succinate, oxaloacetate, or pyruvate) (Huang et al, 2017). In fact, a cell-permeable αKG (dimethyl-αKG) did not rescue the proliferation nor sprouting and senescence in NIT2$^{-/-}$ or NIT2/GLS1$^{-/-}$ cells (data not shown). Considering the larger contribution of GLS1 to αKG pools and the fact that cell-permeable αKG only partially rescues proliferation in glutamine-depleted cells, we looked at other metabolites that were decreased in NIT2$^{-/-}$ cells that might mediate the function of NIT2 under steady state conditions. Interestingly, adenine was the most significantly decreased metabolite in NIT2 knockout cells pointing to alterations in purine handling. Lack of nucleotides decreases proliferation and pushes cells towards senescence (Wiley and Campisi, 2021). To address whether a similar effect is operative in the present model, β-galactosidase staining was performed. Deletion of NIT2 or GLS1 both increased the number of senescent cells, and combined deletion of both enzymes resulted in an additive effect (Fig. 5F). In contrast to this, young endothelial NIT2$^{-/-}$ mice did not exhibit signs of senescence as determined by CellEvent™ (Invitrogen, C10850). In endothelial MACE (Massive Analysis of cDNA Ends)-sequencing from these animals, there was, however, induction of some senescence-associated genes, albeit the effect was rather weak (Fig. EV6).

To determine whether the senescent phenotype in primary endothelial cells was a consequence of a potential shortage of nucleotides, a nucleoside mix (EmbryoMax® Nucleosides, Merck #ES-008) was administered. This reverted the NIT2-deletion-induced senescence phenotype and only attenuated senescence rates in GLS1$^{-/-}$ and double knockout cells (Fig. 5G). Furthermore, we supplemented NIT2$^{-/-}$ cells with adenosine. It decreased the senescent phenotype by 50% (Fig. 5H) and fully rescued the impaired endothelial sprouting in NIT2$^{-/-}$ cells (Fig. 5I,J). These observations suggest that under steady state conditions, the GTωA pathway might couple reactions that are important for the salvage of nucleotides in endothelial cells, a function that is yet not characterized and goes beyond the scope of this study. Altogether, NIT2 and GLS1 are active and maintain endothelial cell proliferation, sprouting, and avert senescence.

## H$_2$O$_2$ inactivates NIT2 by cysteine oxidation

The initial decision to study NIT2 was based on the potential indication that it is inactivated by H$_2$O$_2$. To determine whether this is really the case, we focused on H$_2$O$_2$-dependent cysteine oxidation of the enzyme. We used the biotinylated iodoacetamide (BIAM) switch assay coupled to western blot and proteomics as we previously described (Löwe et al, 2019). Briefly, HUVEC were exposed to 300 μM H$_2$O$_2$ or basal medium for 15 min.

Subsequently, free thiols were blocked with *N*-ethylmaleimide (NEM). Then, reversibly oxidized thiols were reduced with DTT and newly released thiols were labeled with EZ-Link™ Iodoacetyl-PEG2-Biotin (BIAM, Fig. 6A). BIAM-labeled proteins were then enriched with streptavidin-coupled beads, and after electrophoresis and western blotting, samples were probed with anti-NIT2 antibody. Significantly more oxidized NIT2 was pulled down after exposure to H$_2$O$_2$ in comparison to treatment with basal medium (Fig. 6B), demonstrating that NIT2 is oxidized by H$_2$O$_2$. When endothelial cells were treated with H$_2$O$_2$, NIT2 oxidation was dependent on both H$_2$O$_2$ concentration and exposure time, with the largest effect occurring at 300 μM H$_2$O$_2$ and 15-30 min of exposure (Figs. 6C,D and EV7A). To determine whether NIT2 oxidation was restricted to endothelial cells, the BIAM switch assay was repeated using several other cell types, including human carotid and aortic endothelial cells, fibroblasts, smooth muscle cells, and HEK 293 cells. In all of the cell types studied, H$_2$O$_2$ elicited the oxidation of NIT2 (Fig. EV6A–F). However, the oxidation of NIT2 was not induced by other types of ROS as neither diamide (up to 100 μM) nor menadione (up to 50 μM) was able to oxidize NIT2 (Fig. EV7G,H).

NIT2 contains seven cysteine residues with one residing at the catalytic triad (C153-K112-E43, Fig. 6E). Interestingly, NIT2 in vertebrates has higher Cys content than that of other species, and only Cys153 is conserved down to yeast, bacteria, and plants (Fig. EV8). To identify the cysteine residues, which are oxidized by H$_2$O$_2$ we first performed LC-MS of the proteins enriched in the BIAM switch assay. NIT2 could be identified, but with low peptide counts and insufficient coverage. Alternatively, NIT2 was pulled down using an anti-NIT2 antibody instead of the streptavidin antibody, but the NIT2 peptide count was too low for quantitative measurements and not all cysteines were identified. The only successful approach that identified multiple peptides covering six out of seven cysteine residues of NIT2 was based on the overexpression of a His-tagged NIT2 in HEK 293 followed by its purification by affinity chromatography. This approach yielded one peak (peak 2) containing pure NIT2 (Fig. 6F, see Coomassie blue staining). The purified enzyme was subsequently treated without or with H$_2$O$_2$ (300 μM) in the presence of succinamic acid and hydroxylamine for the identification of the redox-sensitive cysteines and activity assay as depicted in Fig. 6F. Using this workflow for LC-MS, the ratio of chloroacetamide (CAM) to N-ethylmaleimide (NEM) labeling should reflect the reversible oxidations induced by H$_2$O$_2$. While the Cys153 was not significantly modified by H$_2$O$_2$, Cys44 and Cys146 showed a two and a threefold significant increase in CAM/NEM ratio, respectively when exposed to H$_2$O$_2$, indicating a reversible oxidation of these cysteine residues (Fig. 6G).

To investigate how Cys modification affected NIT2, we assayed enzyme activity by following the competitive amine substitution of succinamic acid by hydroxylamine, as previously described

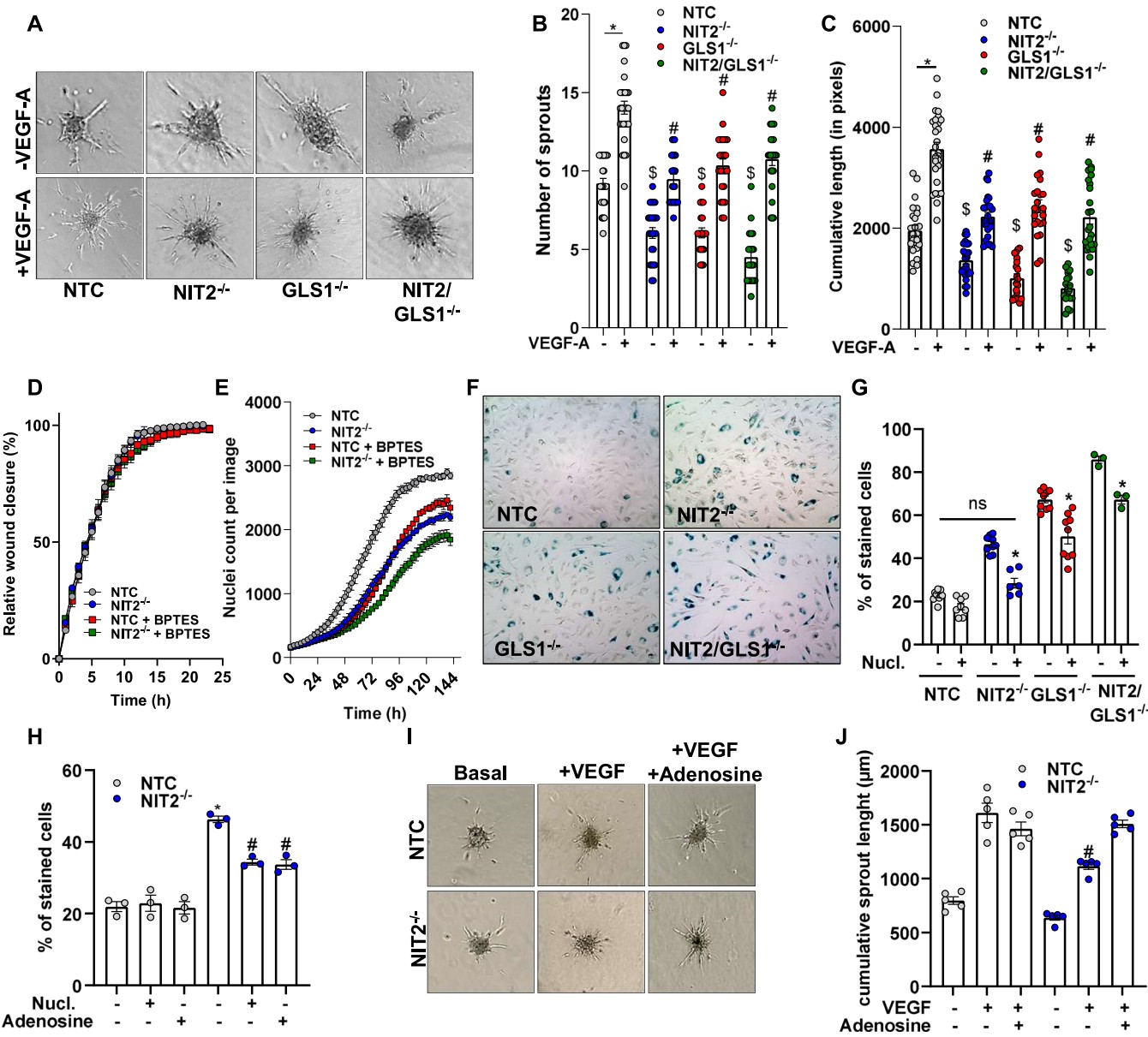

**Figure 5. Deletion of NIT2 results in endothelial cell senescence.**

(A–C) sprout outgrowth assay of HUVEC with and without VEGF-A (10 ng/mL) as indicated. $P < 0.05 as compared to NTC without VEGF-A and #P < 0.05 as compared to NTC with VEGF-A, one-way ANOVA, Bonferroni correction. (D) Migration and (E) proliferation assays. (F–H) Senescence assay with β-galactosidase staining in HUVEC as indicated. Nucl nucleosides. *P < 0.05 as compared to NTC, one-way ANOVA with Bonferroni correction. #P < 0.05 as compared to NIT2$^{-/-}$. (I, J) sprout outgrowth assay of HUVEC with and without VEGF-A (10 ng/mL) and adenosine (8 mg/L) as indicated. #P < 0.05 NTC + VEGF as compared to NIT2$^{-/-}$ + VEGF, one-way ANOVA with Bonferroni correction. Source data are available online for this figure.

(Krasnikov et al, 2009). Incubation of wild-type NIT2 with 300 µM $H_2O_2$ inhibited its activity by 90% whereas a catalytically dead mutant (C153S) was inactive (Fig. 6H). Since Cys44 and Cys146 were reversibly oxidized by $H_2O_2$, we generated mutants of these cysteines. Cys146 is located in a structural loop (Fig. 6E), and replacing it with either serine (C146S) or aspartate (C146D) resulted in low expression and low yield in affinity purification. Replacing the cysteine with alanine (C146A) resulted in an unstable protein, as detected using a thermal shift assay (Fig. EV9A–C). The C146A NIT2 mutant retained 70% of the basal activity of the wild-

type enzyme but remained sensitive to $H_2O_2$. In contrast, the mutation of Cys44 to serine (C44S) resulted in a pronounced loss of activity (Fig. 6H). Cys44 is located in a tunnel that extends from the surface of the protein to the catalytic center, forming the substrate channel (Figs. 6I and EV10A–D). While Cys44 is approximately 11 Å from Cys153, the surrounding negative charge makes it unlikely that these cysteines form a disulfide bond. Therefore, we conclude that Cys44, which forms the substrate channel, is the main redox-sensitive cysteine, and its oxidation inhibits NIT2 activity, probably by hindering substrate access to the active center.

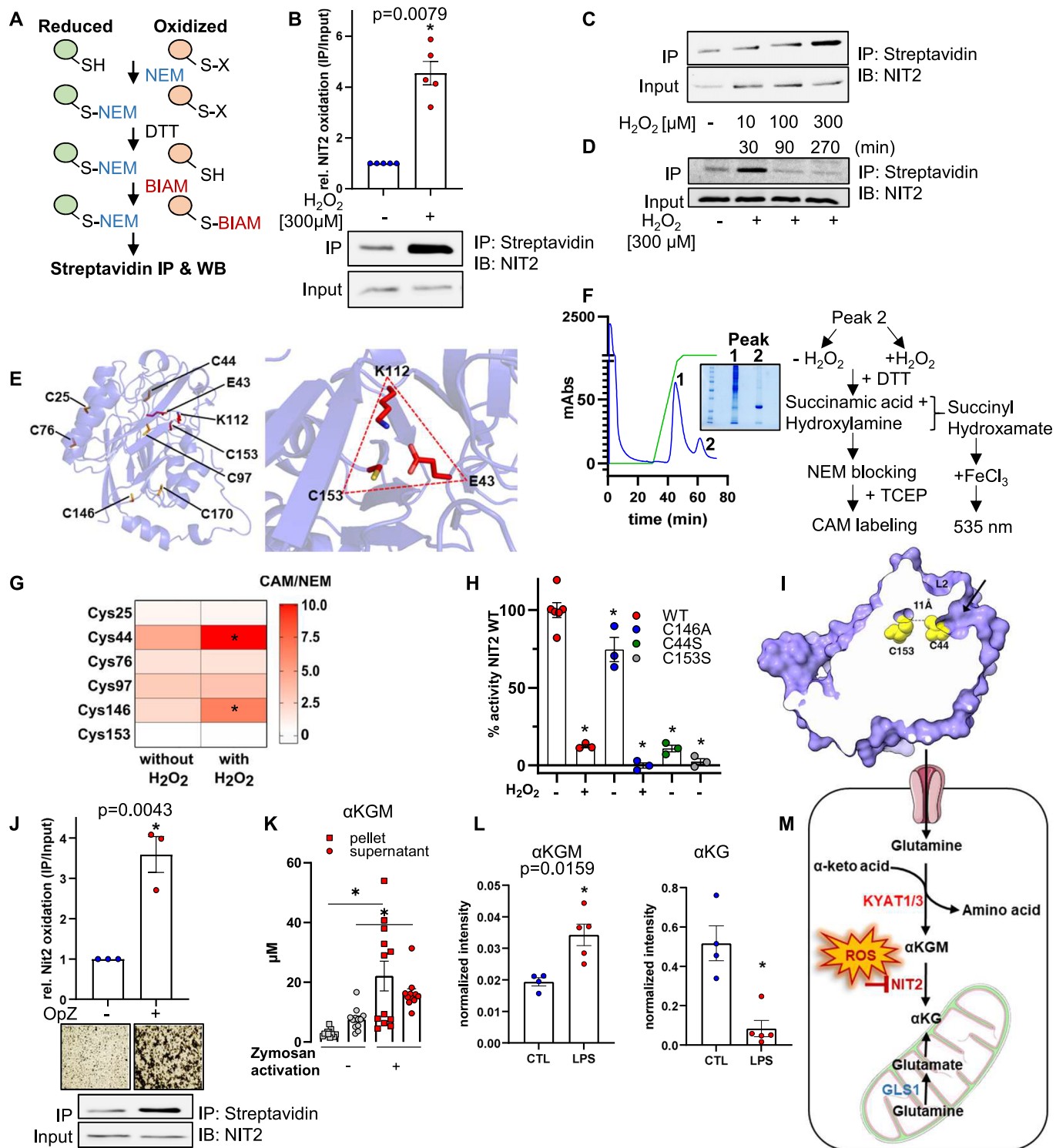

## NIT2 oxidation occurs under pathophysiological conditions

Exogenous application of fairly high concentrations of $H_2O_2$ is not a physiologically important model. We, therefore, set out to detect whether NIT2 oxidation can also be caused in response to physiological stimulation of ROS production. To address this, the

effect of activated granulocytes on endothelial cells was studied: Endothelial cells were co-incubated with human granulocytes, and these were activated with opsonized zymosan to stimulate their NADPH oxidase. Importantly, as previously observed in response to $H_2O_2$, activated granulocytes generated strong oxidation of NIT2 (BIAM, Fig. 6J) and also increased αKGM levels in the pellet and supernatant of HUVEC (LC-MS/MS, Fig. 6K). In the next step, we

**Figure 6.   H₂O₂ inactivates NIT2 by cysteine oxidation.**

(A) Schematic representation of biotinylated iodoacetamide (BIAM) switch assay. (B–D) BIAM switch assay followed by immunoblotting for NIT2 in HUVEC exposed to different concentrations of $H_2O_2$ for different durations. IP immunoprecipitation, IB immunoblotting. (E) Cartoon representation of the predicted structure of human NIT2 (AlphaFold2). The seven cysteine residues are depicted. The insert shows a zoom-in view of Cys153 and the catalytic center (C153-L112-E43). (F) Schematic representation of overexpression, affinity purification, redox LC-MS, and activity assay for NIT2. (G) Heatmap summarizing reversible modifications of Cys in NIT2. (H) NIT2 activity assay using competitive amine substitution of succinamic acid by hydroxylamine. $*P < 0.05$ as compared to wild-type NIT2 without $H_2O_2$, one-way ANOVA with Bonferroni correction. (I) Solvent-exposed surface of the predicted NIT2 model depicted in purple. Cys153 and Cys44 (yellow spheres) are 11 Å apart. Cys44 localizes at the bottom of the open substrate binding channel, capped by the flexible loop 2 (L2). Arrow shows a tunnel from the protein surface down to the Cys153. (J) BIAM switch assay followed by immunoblotting for NIT2 and αKGM measurement (K) in HUVEC co-incubated with human granulocytes with or without zymosan opsonized by human plasma. $*P < 0.05$, without vs. with stimulation, t test. Microscopy images (J) show aggregation of granulocytes in response to zymosan. Targeted LC-MS/MS for αKGM and αKG in urine (L) of mice treated with lipopolysaccharide (LPS, 4 mg/kg, 4 h). (M) NIT2 is a redox switch in the glutamine transaminase-ω-amidase pathway. Source data are available online for this figure.

sought for in vivo evidence for NIT2 oxidation. Given that NIT2 is positioned between αKGM and αKG, an increase in the ratio of the two metabolites would reflect loss of enzyme activity. A classic inflammatory and high ROS model, LPS injection (LPS, 4 mg/kg, 4 h) was studied in mice, and analyses were performed in the urine, postulating a high clearance of the lactam molecule αKGM as a metabolic end product. Indeed, after LPS injection, urinary levels of αKGM significantly increased and levels of αKG decreased (Fig. 6L). With respect to the ratio, this increased from 0.07 to 0.4, which is more than tenfold. Thus, NIT2 oxidation not only occurs in response to exogenous application of $H_2O_2$, but also occurs in response to cellular activation as well as in vivo.

# Discussion

In this study, we aimed to identify metabolic targets of oxidative stress in endothelial cells. For this, we exposed HUVEC to menadione or $H_2O_2$ as oxidizing agents that have different properties. Exposure of endothelial cells to $H_2O_2$ selectively unmasked a metabolic pathway for glutamine that has been largely overlooked and underappreciated. GLS1 within the glutaminase I pathway was hitherto the only known enzyme of glutamine metabolism contributing to endothelial αKG production. Here, we present the glutamine transaminase-ω-amidase (GTωA) pathway that is a redox switch in glutamine metabolism and is important for non-canonical generation of αKG and angiogenesis.

The GTωA pathway consists of two coupled reactions: (i) the transamination of glutamine by the enzymes KYAT1 and KYAT3 to generate αKGM. This reaction is accompanied by the replenishment of amino acids using their α-keto acids as substrates. (ii) αKGM is converted to αKG by NIT2. Exposure of endothelial cells to $H_2O_2$ increased αKGM and conversely decreased αKG, suggesting that the reaction catalyzed by NIT2 is redox-sensitive. Using the BIAM switch assay, redox proteomics, and site-directed mutagenesis, we demonstrated that NIT2 is reversibly oxidized by $H_2O_2$ and that Cys44 and Cys146 are the reactive cysteines. The lack of a crystal structure for human NIT2 with the substrate bound at its active site limited our ability to model how oxidation of Cys44 and Cys146 affects conformational changes, substrate binding, and multimerization that may be important for NIT2 activity.

Cys146 is located in a surface loop and scores the highest with algorithms that predict cysteine oxidation (Keßler et al, 2021). This cysteine seems to be important for proper folding or stability of the protein, as the C146S and C146D mutants were unstable. In contrast, Cys44 was directly linked to the catalytic activity of NIT2. Despite the high confidence score for the modeling (>90) with AlphaFold2 for human NIT2, its crystal structure is not available, and the mouse structure (Barglow et al, 2008) does not contain the substrate bound to it what limits the understanding of the role of Cys44 in NIT2 catalysis. The variant C44S abolished NIT2 activity, suggesting that given its localization in the open substrate tunnel, Cys44 is the primary redox switch and it might have an important role in the initiation of the catalytic cycle.

How relevant is NIT2 for endothelial metabolism? To address this question, we first looked at αKG, which is a major metabolic hub. Knockdown of GLS1 decreases αKG by 40% (Kim et al, 2017), so that alternative pathways such as the GTωA might be relevant. Measurements of cellular αKG content and isotopic tracing of glutamine suggested that the GLS1 pathway is dominant for αKG and might override the contribution of NIT2 to metabolic αKG pools under basal conditions. When NIT2 activity is lower either by acute oxidative inhibition (in the present study through exposure to $H_2O_2$), CRISPR/cas9 deletion or lower expression, αKGM accumulates and no longer serves as substrate for NIT2 to generate αKG, unmasking the GTωA pathway as a salvage for αKG. In line with that, a double deletion of GLS1 and NIT2 showed a more profound decrease in αKG pools and had a larger functional impact on endothelial cell sprouting, proliferation, and senescence.

The in vivo importance of endothelial NIT2 was studied by generating ecNit2$^{-/-}$ mice. Endothelial-specific deletion of NIT2 led to an accumulation of αKGM in plasma, whereas αKG levels were conversely decreased. Given that all cells, but particularly the liver and the kidney, are thought to contribute to plasma αKG, the latter finding is unexpected. It demonstrates that in vivo under normal conditions, the relevance of endothelial NIT2 might be greater than what is suggested by the cell culture studies using CRISPR/Cas9-mediated deletion.

Functionally, the knockout of NIT2 in endothelial cells had similar effects to that of the knockout of GLS1 (Kim et al, 2017; Huang et al, 2017): it did not affect migration but decreased proliferation and decreased angiogenesis in endothelial cells and in the mouse retina. Moreover, deletion of NIT2 resulted in premature senescence of endothelial cells in culture. Double knockout of NIT2 and GLS1 (NIT2/GLS1$^{-/-}$) potentiated all these effects. The question that therefore arises is why endothelial cells contain both pathways? The GTωA may be advantageous for endothelial cells as the production of αKG from glutamine does not involve a net oxidation and thus does not regenerate NAD(P)H as in the glutaminase I pathway. In this way, αKG production can be

uncoupled from mitochondria. Indeed, we observed by immuno-fluorescence that NIT2 is distributed across endothelial cells and not restricted to the mitochondria as GLS1 (Kim et al, 2017). Another potential metabolic advantage of GTωA pathway might be the salvage of α-keto acids to their corresponding amino acids (Fig. 6M) and coupling of reactions that salvage nucleotides like adenine. Interestingly, the addition of adenosine to NIT2$^{−/−}$ cells partially rescued senescence and fully restored endothelial sprouting, suggesting a function for NIT2 beyond the generation of αKG. The link between NIT2 and adenine, however, is unclear from the current metabolic pathways. It is surprising that the GTωA pathway has so far not gained much attention. Potentially, this is a consequence of a lack of standard, but also of the short half-life of αKGM in its linear form and its effective de-amidation to αKG by NIT2. αKGM (lactam form) levels in cells are essentially undetectable in the presence of active NIT2, and plasma levels are lower than αKG, as demonstrated in the present study. αKGM was previously produced through oxidation of L-glutamine with snake venom L-amino acid oxidase in the presence of catalase (Shen et al, 2020). αKGM measurements in its lactam form were originally performed by gas chromatography or HPLC(Shurubor et al, 2016). Such methods have drawbacks, but set the basis for the first characterization of the GTωA pathway under normal and pathological conditions. For example, αKGM is elevated in the cerebrospinal fluid of hyperammonemic patients with hepatic encephalopathy and in the urine of hyperammonemic patients with an inborn error of the urea cycle or citrin deficiency (Vergara et al, 1974; Duffy et al, 1974a; Duffy et al, 1974b; Cooper and Kuhara, 2014; Kuhara et al, 2011).

The detection and semi-quantification of αKGM in untargeted metabolomics was based on the predicted fragmentation pattern. The identity of αKGM was validated for the first time in this study via LC-MS/MS. Importantly, data from untargeted metabolomics that contain αKGM (Chen et al, 2023) allowed for a co-localization analysis that found an association of a NIT2 variant with higher αKGM in plasma. In addition, the increasing number of publicly available metabolomic datasets provides opportunities to mine data for associations between αKGM and diseases, and to further evaluate the functions of the GTωA pathway. This applies in particular to cells and tissues with high glutamine utilization, such as cancer cells, the liver, and the kidneys. From the metabolomics datasets available, αKGM is one out of nine significantly increased metabolites in the serum of patients with chronic and large ischemic infarction volume (Sidorov et al, 2021). In patients with chronic kidney disease, αKGM is among the five significantly increased metabolites with the strongest associations with urinary uromodulin (Bächle et al, 2023). Inasmuch as αKGM is increased in the urine of LPS-treated mice, it is possible that this metabolite is a marker for oxidative stress in pro-inflammatory conditions as a result of NIT2 inhibition under oxidative conditions.

In conclusion, we present the ω-amidase/NIT2 as a molecular target of oxidative stress and a redox switch in metabolism. Its inhibition by oxidation at cysteine residues leads to an accumulation of αKGM, a metabolite that was previously not fully studied. Our work provides the first systematic analysis of the role of NIT2 in the GTωA pathway, which serves as an alternative and non-canonical route for the generation of αKG from glutamine in endothelial cells. Deletion of NIT2 decreased endothelial cell proliferation, angiogenesis, and induced senescence. We focused on

the endothelial function of NIT2 due to the great dependence of endothelial cells on glutamine. It could be inferred that the GTωA pathway is equally important in other cell types and organs with high glutamine utilization, such as cancer cells, the liver, and the kidneys. Thus, our findings in EC may inspire future research on glutamine metabolism and the biological importance of αKGM and the GTωA pathway.

# Methods

**Reagents and tools table**

| Reagent/resource | Reference or source | Identifier or catalog number |
|---|---|---|
| **Experimental models** | | |
| Human umbilical vein endothelial cells (HUVEC) | PromoCell | #C12203 |
| Human embryonic kidney 293 cells (HEK 293) | ATCC | #CRL-1573 |
| Lenti-X 293T cells | Takara | #632180 |
| **Recombinant DNA** | | |
| 10x His-Tag NIT2 plasmid | Sino Biological | #HG23517-CH |
| Q5® Site-Directed Mutagenesis Kit | New England Bio Labs | #E0554S |
| CRISPR/Cas9 v2 (LCV2) plasmid puromycin | Addgene plasmid | #52961 |
| CRISPR/Cas9 v2 (LCV2) plasmid hygromycin | Frank Schnütgen, Department, Goethe University | |
| pMD2.G plasmid | Addgene | #12259 |
| psPAX2 plasmid | Addgene | #12260 |
| **Antibodies** | | |
| β-Actin | Sigma-Aldrich | #A1978 |
| GLS1 | Abcam | #ab156876 |
| NIT2 | Abcam | #ab183074 |
| Anti-His-tag | Bethyl | #A190-114A |
| CD31 | R&D Systems | #AF3628 |
| **Oligonucleotides and other sequence-based reagents** | | |
| PCR primers | This study | listed in the methods section |
| Cas9-GFP, Alt-R CRISPR-Cas9 sgRNA, Alt-R HDR Donor Oligo | IDT | #10008161 |
| RNA Mini Kit | Bio&Sell | # BS67.311 |
| **Chemicals, enzymes, and other reagents** | | |
| α-ketoglutaramate | This study | |
| Endothelial basal medium | PeloBiotech | #PB-C-MH-100-2199 |
| Dulbecco's Modified Eagle Medium High Glucose | Gibco | #15140-122 |
| Medium 199 | Sigma | #M0650 |

| Reagent/resource | Reference or source | Identifier or catalog number |
|---|---|---|
| $^{13}C_5$-$^{15}N_2$-L-Glutamine | Cambridge Isotopes Laboratories | #CNLM-1275-H-0.1 |
| Alt-R Cas9 Electroporation Enhancer | IDT | #1075915 |
| SuperScript III Reverse Transcriptase | ThermoFisher Scientific | #12574026 |
| SYBR Green Supermix and ROX | Bio-Rad | #1725125 |
| ProteaseMAX | Thermo Scientific | # A40007 |
| Isolectin B4 | ThermoFisher | #I21411 |
| Rat-tail collagen | BD | #354236 |
| β-Galactosidase Staining Kit | Cell Signaling Technology | #9860 |
| **Software** | | |
| R | https://ropensci.org/ | |
| SNEEP | v1.0, https://doi.org/10.5281/zenodo.10830008 | |
| EpiRegio | https://doi.org/10.1093/nar/gkaa382, REM-gene interactions | |
| CRISPOR | http://crispor.tefor.net | |
| ZEN lite 3.1 | Carl Zeiss Microscopy | |
| Angiotool64 (version 0.6a) | https://ccrod.cancer.gov/confluence/display/ROB2/Home | |
| Prism 9.2.0. | GraphPad | |
| **Other** | | |
| QTrap 5500 LC-MS/MS mass spectrometer | Sciex | |
| 6500 + ESI-tripleQ MS/MS | Sciex | |
| AriaMX cycler | Agilent | |
| HisTrap FF column packed with Ni-sepharose | Cytiva | #17531901 |
| Äkta FPLC system | GE Healthcare/Cytiva | |
| Q Exactive Plus Orbitrap equipped with an UHPLC Dionex Ultimate 3000 instrument | ThermoFisher Scientific | |
| Odyssey | Licor | |
| Phoenix MICRON IV Image-Guided Laser System | Phoenix MICRON | |
| NextSeq2000 instrument | Illumina | |

Protocols will be fulfilled by Dr. Flávia Rezende (rezende@vrc.uni-frankfurt.de). Mouse lines will be shared under a Materials Transfer Agreement. Results from untargeted metabolomics and RNAseq are available in the supplementary material.

## Cell culture

Pooled human umbilical vein endothelial cells (HUVEC) were purchased from PromoCell (#C12203, Heidelberg, Germany) and cultured in endothelial growth medium (EGM), consisting of endothelial basal medium (EBM) supplemented with 8% fetal calf serum, 0.5% penicillin/streptomycin (50 µg/mL), growth factors (EGF, bFGF, IGF, VEGF, #PB-C-MH-100-2199, PeloBiotech, Germany), heparin, L-glutamine but without hydrocortisone. For each experiment, at least three different batches of HUVEC at the highest passage number 4 were used. Human embryonic kidney 293 cells (HEK 293) (ATCC, Manassas, USA) and Lenti-X 293T cells (Takara, 632180, Japan) were cultured in Dulbecco's Modified Eagle Medium High Glucose (Gibco) supplemented with 8% FCS, penicillin/streptomycin (50 µg/mL of each) (#15140-122, Gibco/Lifetechnologies, USA). All cells were cultured in a humidified atmosphere (5% $CO_2$, 37 °C).

## Untargeted metabolomics and data analysis

Untargeted metabolomics of HUVEC exposed to menadione (5 µM) or $H_2O_2$ (300 µM) has been previously published (Müller et al, 2022). Untargeted metabolomics of CRISPR/cas9 NIT2$^{-/-}$, GLS1$^{-/-}$, and NIT2/GLS1$^{-/-}$ HUVEC was performed from cells grown in EGM as described (Müller et al, 2022).

Batch-normalized intensity data were analyzed for differential metabolite levels in R (https://ropensci.org/) using the metabodiff (v0.9.5) package. The data were subjected to k-nearest neighbor imputation, with a cutoff value of 0.25. Further normalization was performed using variance-stabilizing normalization, followed by quality checks to ensure no artefacts were introduced by the procedure. Corresponding treatment and control samples were then compared using the differential test function, which implements a Student's $T$ Test followed by Benjamini–Hochberg correction for multiple testing. Results were visualized using ggplot2 (v3.3.5, RRID:SCR_014601).

## Establishment of αKGM as LC-MS/MS standard

Pure αKGM was synthesized as previously described (Shen et al, 2020). The heavy isotopologue (m + 5 for carbons and m + 1 for nitrogen) of αKGM was synthesized from [$^{13}C_5$-$^{15}N_2$]-L-Glutamine (Cambridge Isotopes Laboratories Inc., Tewksbury, USA). The fragmentation of αKGM was evaluated on a QTrap 5500 LC-MS/MS mass spectrometer (Sciex) via direct injection. Different collision energies (CE) were first tested in negative ionization mode, and a CE of 40 gave the best fragmentation pattern of m/z 144 Da (parental ion) as m/z 126 Da (most abundant), 82 Da (2nd most abundant), and 42 Da (3rd most abundant). Alternatively, αKGM was measured in positive ionization mode using a Sciex 6500 + ESI-tripleQ MS/MS on low mass mode (0–1000 Da) with declustering potential, exit potential, collision energy, and collision cell exit potential of 206, 10, 21, 12 and 206, 10, 25, and 14 Volts, respectively. The dwell time was 20 milliseconds. Two fragments of the parental ion of αKGM (m/z 147 Da) showed m/z 105.1 Da and m/z 91 Da.

## Metabolite quantitative trait locus (mQTL) in NIT2

To evaluate whether the same genetic variants are driving the associations with circulating αKGM levels and NIT2 transcript levels, a co-localization analysis was employed. The genetic associations for plasma αKGM nearby NIT2 coding region were obtained from Chen et al (2023).(Chen et al, 2023) Their genome-wide association study

(GWAS) was conducted using the genetic data and metabolite measurements of individuals in European ancestry from the CLSA (Canadian Longitudinal Study on Aging) cohort ($n$ = 8203). The *cis*-eQTL summary statistics for NIT2 in artery tibial and artery aorta were obtained from the GTEx v8 ($n$ = 584 for artery tibial and $n$ = 387 for artery aorta) (GTEx Consortium 2020). Specifically, we applied a stringent Bayesian method using coloc R package (5.1.0) with the priors recommended by the original study ($P1 = 1 \times 10^{-4}$, $P2 = 1 \times 10^{-4}$, $P12 = 1 \times 10^{-5}$) (Giambartolomei et al, 2014) to estimate the posterior probabilities (PP) that the αKGM and the NIT2 transcript share a single causal genetic locus. The genetic variants in the ±500 kb window of the leading metabolite quantitative trait locus (mQTL) of αKGM, rs3830303, that has minor allele frequency (MAF) over 0.05 were used for the analysis. The PP that two traits share one causal SNV (single-nucleotide variant) above 80%, namely PP.H4 > 0.8, was considered to be colocalized. LocusZoom plots for the tested genetic region were plotted using locuszoomr R package (0.1.3). The linkage distribution (LD) $r^2$ for the genetic variants was obtained from the genetic data of European individuals in the 1000 Genome phase 3 reference panel (Auton et al, 2015). Genetic variants were plotted based on their LD to the leading mQTL. The αKGM GWAS from individuals of European ancestry can be found on the GWAS Catalog (https://www.ebi.ac.uk/gwas/) with accession number (GCST90199885). GTEx V8 release of eQTL data from individuals of European ancestry can be found in https://www.gtexportal.org/home/datasets.

### rs277627 analysis

To identify whether the SNV rs277627 is regulatory, we applied SNEEP (v1.0, https://doi.org/10.5281/zenodo.10830008), a statistical approach to identify whether a SNV significantly affects a TF binding site (Baumgarten et al, 2024). For rs277627 (chr3: 100,336,428–100,336,429, hg38), we compared the risk allele A against the major allele G. SNEEP requires as input a TF motif set, for which we used 817 non-redundant human motifs from JASPAR (version 2022) (https://doi.org/10.1093/nar/gkab1113), HOCO-MOCO (https://doi.org/10.1093/nar/gkx1106) and the Kellis ENCODE motif database (https://doi.org/10.1093/nar/gkt1249). To link the SNV to potential target genes, we downloaded 2.4 million regulatory elements (REM) associated to target genes, which are publicly available at the EpiRegio webserver (https://doi.org/10.1093/nar/gkaa382, REM-gene interactions: https://doi.org/10.5281/zenodo.3750929, filename: REMAnnotation.txt). The following parameters were specified to run SNEEP: -p 0.5, -c 0.01, -r REMAnnotation.txt, -g ensemblID_GeneName.txt. The SNEEP software, considered TF motifs, the file ensemblID_GeneName.txt and more details about the specified parameters are available at the GitHub repository (https://github.com/SchulzLab/SNEEP).

### Deletion of the regulatory element (REM) in the NIT2 gene

The REM that contains the SNV rs277627 was deleted by CRISPR/cas9 in HEK 293 cells using a dual gRNA approach (see CRISPR/cas9 section). After viral transduction, cells were clonally expanded by serial dilution, and one single clone was obtained. Validation of the CRISPR/Cas9 knockout of the target REM was performed by

genomic PCR and Sanger sequencing from genomic DNA. CRISPR/Cas9 target sites were amplified by PCR (forward primer: ATCAGAGGCCAGGGTTTGTC, reverse primer: GAGGAAGG-CAGGCATCTGTC) with PCR Mastermix (ThermoFisher, #K0171) and 500 ng DNA followed by agarose gel electrophoresis and ethidium bromide staining. Bands were excised from the gel and subjected to Sanger sequencing to further validate a deletion of 437 bp.

### CRISPR/Cas9-mediated NIT2 SNP mutation via homology-directed repair

For genome editing of the NIT2 SNP (rs277627) located on chr3:100336429 (G-A, hg38), 100,000 HUVEC were electroporated in E2 buffer with the NEON electroporation system (Invitrogen) (1400 V, 1 × 30 ms pulse) with Cas9-GFP (#10008161, Alt-R S.p. Cas9-GFP V3, IDT, Belgium), an Alt-R CRISPR-Cas9 sgRNA (IDT, Belgium) and a SNP-containing homology-directed repair (HDR) donor template (Alt-R HDR Donor Oligo, IDT, Belgium) according to the manufacturer's protocol (IDT, Belgium) with Alt-R Cas9 Electroporation Enhancer (#1075915, IDT, Belgium). The sgRNA and donor oligo sequence were designed with the Alt-R HDR Design Tool (IDT, Belgium). The sgRNA used to target the NIT2 intron locus had a cut-to-mutation distance of 0 and the following sequence: 5'-GTA TAG AGA AAG AAG TAG GG-3'. The following HDR donor oligo (length 81nt, left arm 40nt, right arm 40nt) (IDT, Belgium) was used: 5'-GAA AAG AAA AAG ACA CAA ACA AAG TAT AGA GAA AGA AGT AAG GGG GCC CAG GGG ACC AGC GTT CGG CAT ACG GAG GAT CCC-3'. A full medium exchange was done every 24 h, and cells were incubated for 7 days. Alt-R HDR Enhancer V2 (#10007921, IDT, Belgium) was used to enhance efficiency. Afterward, cells were split for genomic DNA and RNA isolation. The NIT2 SNP insertion was verified from genomic DNA by PCR (forward primer: 5'-CCT GAG CCA AAC AGG CCT TC-3', reverse primer: 5'-TCC TAT GAC CCT GCC ACA TC-3', leading to a PCR product of 391nt). The PCR product cut out, purified and sequenced with Sanger-Seq with the reverse primer (5'-TCC TAT GAC CCT GCC ACA TC-3).

### RT-qPCR

Total RNA isolation was performed with the RNA Mini Kit (Bio&Sell) according to the manufacturer's protocol, and reverse transcription was performed with the SuperScript III Reverse Transcriptase (#12574026, ThermoFisher Scientific, MA, USA) using a combination of oligo(dT)23 and random hexamer primers (Sigma). The resulting cDNA was amplified in an AriaMX cycler (Agilent) with the ITaq Universal SYBR Green Supermix and ROX as reference dye (Bio-Rad, #1725125). Forward primer: GGTGCTTGTCTGCAGAGTCAT, reverse primer: TCTTCAGG-GATAGAGCCTCCAA for *NIT2*. Relative expression was calculated using the ΔΔCt method, and genes were normalized to β-actin.

### Biotinylated iodoacetamide (BIAM) switch assay

HUVEC were exposed to either $H_2O_2$ (300 μM), menadione (5 or 50 μM) or diamide (100 or 500 μM) in EBM in a time course manner. Thiols were blocked with N-ethylmaleimide (NEM,

100 mM), and the BIAM switch assay was performed as previously described (Löwe et al, 2019).

## Expression and purification of recombinant NIT2

The human 10x His-Tag NIT2 plasmid was obtained from Sino Biological (#HG23517-CH, Wayne, USA). NIT2 mutants were generated by site-directed mutagenesis to obtain NIT2-C146S, C146D, C146A, and C44S (Q5® Site-Directed Mutagenesis Kit, New England Bio Labs # E0554S).

Plasmid was transiently overexpressed for 48 h in HEK 293 cells using PEI (polyethyleneimine, DNA: PEI ratio 1:5) for transfection. Subsequently, cells were lysed in Triton X-100 buffer supplemented with protease inhibitors. The extract was loaded on a HisTrap FF column packed with Ni-sepharose (Cytiva, #17531901) and purified using an Äkta FPLC system (GE Healthcare/ Cytiva, Solingen, Germany) with a flow rate of 0.5 mL/min. Proteins were eluted with a linear gradient of imidazole up to 500 mM.

## Thermal shift assay

Thermal shift assay was performed as previously described (Niesen et al, 2007; Kramer et al, 2019). Briefly, differential scanning fluorimetry was performed in a PCR plate with a total volume of 40 μL. Purified NIT2-WT ($C_{final}$ 5 μM) or NIT2-C146A ($C_{final}$ 5 μM), Triton X-100 (0.001% w/v), and SYPRO Orange (Thermo-Fisher Scientific) (2.5×) were mixed in phosphate buffer with or without DTT (100 mM KPO$_4$, 5 mM DTT). The samples were measured on an Icycler IQ single-color real-time PCR system ($\lambda_{ex} = 490$ nm, $\lambda_{em} = 570$ nm) and emission was recorded during a temperature gradient of 0.2 °C increase per 24 s (25–80 °C). Raw data from both, NIT2-WT and NIT2-C146A, measurements were analyzed directly using a Boltzmann sigmoidal fit in GraphPad Prism. The $V_{50}$ values were considered as the melting temperatures. All samples were measured in triplicate.

## NIT2 activity assay

To measure NIT2 activity, the wild-type and mutants of NIT2 were treated with H$_2$O$_2$ (300 μM) for ten minutes in 100 mM KPO$_4$ buffer, pH 7.4. Next, samples were incubated with 5 mM DTT for 10 min followed by the addition of 20 mM succinamic acid as substrate plus 100 mM neutralized hydroxylamine-HCl for 30 min at 37 °C. One aliquot of this NIT2 reaction mixture was used for activity assay, and another for redox proteomics. The hydroxaminolysis activity assay for NIT2 was performed as previously described (Krasnikov et al, 2009).

## Redox proteomics

Free thiols in NIT2 were blocked with 250 mM NEM for 20 min and the protein was precipitated with cold acetone overnight at −20 °C. After centrifugation and acetone evaporation, the resulting pellets were solubilized in 6 M guanidinium chloride (GdmCl) and 10 mM tris(2-carboxyethyl)-phosphine (TCEP, freshly added). Samples were further supplemented with 40 mM chloroacetamide, 1 mM CaCl$_2$, and 0.01% ProteaseMAX (Thermo Scientific™ # A40007). Proteins were digested with 1 μg trypsin (sequencing grade, Promega) overnight at 37 °C. The digestion was stopped with

1% trifluoroacetic acid. Purification and elution of peptides was performed as previously described (Rappsilber et al, 2007). After drying, the peptides were resuspended in 10 μl of 1% acetonitrile, 0.1% formic acid, and stored at −20 °C until MS analysis.

The peptides of each fraction (3 μL) were injected and analyzed by LC-MS/MS using a Q Exactive Plus Orbitrap equipped with an UHPLC Dionex Ultimate 3000 instrument (ThermoFisher Scientific). The peptides were loaded on an Acclaim™ PepMap™ 100 C18 LC Pre-column (0.1 mm × 20 mm, nanoViper, 5 μm, 100 Å) and separated using emitter columns (15 cm length × 100 μm ID × 360 μm OD × 15 μm orifice tip; MS Wil/CoAnn Technologies) filled with ReproSilPur C18-AQ reverse-phase beads of 3 μm, 100 Å (Dr. Maisch GmbH). HPLC settings: linear gradients of 4–25% acetonitrile (ACN), 0.1% formic acid (FA) for 35 min followed by 25–50% ACN, 0.1% FA for 5 min, 50–99% ACN, 0.1% FA for 1 min. The column was washed with 99% ACN, 0.1% FA for 5 min and then equilibrated with 4% ACN, 0.1% FA for 14 min. All flow rates were set as 300 nl·min$^{-1}$. MS data were recorded by DDA. The full MS scan range was 300– 2000 $m/z$ with a resolution of 70,000 and an AGC of 3E6 with a maximal injection time of 65 ms. The 20 most abundant precursors were selected for MS2. Only charged ions >2 and <8 were selected for MS/MS scans with a resolution of 17,500; isolation window of 2.0 $m/z$; AGC: 1E5; maximal injection time: 65 ms. MS data were acquired in profile mode.

MaxQuant 2.0.3.0 (Tyanova et al, 2016) was used to analyze the MS raw spectra files. The human reference proteome database (UniProt, February 2024, including canonical sequences and isoforms) was used for identification with a false discovery rate (FDR) ≤ 1%. To quantify the protein abundances, iBAQ values were calculated. To account for protein loading and MS sensitivity variations, the intensities of individual peptides were normalized using the respective iBAQ values of NIT2 from the respective samples.

## Isolation of granulocytes

Neutrophil granulocytes were isolated as previously described (Kuhns et al, 2015). Zymosan was opsonized with fresh human serum (20 mg/mL) by incubation at 37 °C for 30 min. 20.000.000 granulocytes were directly added over to the HUVEC and subsequently activated with opsonized zymosan (3 mg/mL) for 15 min at 37 °C.

## CRISPR/Cas9 deletions in HUVEC

Guide RNAs targeting coding sequences for NIT2 and GLS1 were designed using the publicly available CRISPR algorithm (www.benchling.com). gRNAs targeting the regulatory element within the *NIT2* gene were designed using CRISPOR (http://crispor.tefor.net) (Haeussler et al, 2016). gRNAs were cloned into a lentiviral CRISPR/Cas9 v2 (LCV2) plasmid using the "Golden Gate" cloning protocol (Sanjana et al, 2014). gRNAs were cloned into plasmids containing either puromycin resistance (gift from Feng Zhang, Addgene plasmid #52961; http://n2t.net/addgene:52961; RRID:Addgene_52961) or hygromycin resistance (kindly provided by Frank Schnütgen, Department of Medicine, Hematology/Oncology, University Hospital Frankfurt, Goethe University, Frankfurt, Germany). Lentiviruses were produced in Lenti-X 293T cells (Takara, #632180) using Polyethylenamine,

psPAX2, and pVSVG (pMD2.G). pMD2.G was a gift from Didier Trono (Addgene plasmid #12259; http://n2t.net/addgene: 12259; RRID:Addgene_12259). psPAX2 was a gift from Didier Trono (Addgene plasmid #12260; http://n2t.net/addge ne:12260; RRI-D:Addgene_12260). LentiCRISPRv2-produced virus was transduced for 24 h in HEK 293 cells or HUVEC (at passage 1) with polybrene transfection reagent (MerckMillipore, #TR-1003-G), and selection was performed with puromycin (1 µg/mL) or hygromycin (100 µg/mL) for 6 days.

List of gRNAs including overhangs

| Gene | Sense gRNA sequence (5´–3´) | Antisense gRNA sequence (5´–3´) |
|------|------|------|
| GLS1_gRNA | CACCGCATCATACCCATAACATTG | AAACCAATGTTATGGGTATGATGC |
| NIT2_gRNA | CACCGGCAGCATATATCTCATTGG | AAACCCAATGAGATATATGCTGCC |
| NTC | CACCGTTCCGGGCTAACAAGTCCT | AAACAGGACTTGTTAGCCCGGAAC |
| REM_gRNA1 | CACCGAGAGATAGCTCTCTAATGGT | AAACACCATTAGAGAGCTATCTCTC |
| REM_gRNA2 | CACCGAAACCGCCTTAGGGCTGGAA | AAACTTCCAGCCCTAAGGCGGTTTC |

## Western blot analysis

HUVEC were lysed with a triton-based buffer at pH 7.4, with the following concentrations in mmol/L: Tris-HCl (50), NaCl (150), sodium pyrophosphate (10), sodium fluoride (20), Triton X-100 (1%), sodium desoxycholate (0.5%), proteinase inhibitor mix, phenylmethylsulfonyl fluoride (1), orthovanadate (2), okadaic acid (0.00001). Proteins (30 µg) were separated by SDS/PAGE, transferred by western blot, and probed with antibodies as listed below. Western blot analyses were performed with an infrared-based detection system (Odyssey, Licor, Bad Homburg, Germany).

List of antibodies

| Name | Host | Manufacturer | ID |
|------|------|------|------|
| β-Actin | Mouse | Sigma-Aldrich, Taufkirchen, Germany | A1978 |
| GLS 1 | Rabbit | Abcam, Cambridge, UK | ab156876 |
| NIT2 | Rabbit | Abcam, Cambridge, UK | ab183074 |
| Anti-His-tag | Rabbit | Bethyl, Texas, USA | A190-114A |

## Immunofluorescence

HUVEC were seeded on Ibidi slides and fixed with 4% paraformaldehyde that was quenched with glycine (2%). Next, cells were permeabilized with 0.05% Triton X-100 in PBS. After blocking with 3% BSA for 30 min, the cells were incubated at 4 °C overnight with a 1:200 dilution of the primary antibody. Cells were washed with 0.3% Tween 20 in PBS and incubated with a 1:500 dilution of secondary antibody for 30 min. The cells were then washed again with 0.3% Tween 20 and counterstained with DAPI (1:500). Images were captured with a laser confocal microscope LSM800, and analyzed with ZEN lite software.

## Targeted LC-MS/MS analysis for TCA cycle metabolites

### Metabolite extraction from cells

HUVEC were grown in EGM on a 10 cm dish until confluence. For isotopic labeling experiments, glutamine was replaced with fully labeled glutamine ($^{13}C_5$, $^{15}N_2$) for 24 h. Cells were washed with ice-cold PBS and scratched carefully in PBS. After centrifugation (2400 × g, 4 min), the cell pellets were lysed by adding 600 µL of 90% methanol and two freeze-thaw cycles. After centrifugation, 300 µL of clear supernatant were mixed with internal standards mix containing eight $^{13}C$-labeled metabolites: α-ketoglutaric acid-1$^{13}C$, citric acid-2-$^{13}C$, glucose-6-phosphate-6-$^{13}C$, D-glucose-1-$^{13}C$, L-glutamic acid-2-$^{13}C$, pyruvate-1-$^{13}C$, succinic acid-2-$^{13}C$ and itaconic acid-5-$^{13}C$. Samples were dried and reconstituted in water containing 0.5% formic acid.

### Metabolite extraction from plasma and urine

Plasma and urine were diluted with 25 µL Trifluorethanol (TFE):$H_2O$ (2:1) or 25 µL of pure TFE, respectively. After a 10 min incubation, samples received 100 µL of MeOH:EtOH (1:1), followed by 100 µL $H_2O$ + 10 µL internal standard mix (containing internal Standards homotaurine, citrate-1,5-$^{13}C_2$ and succinate-1,4-$^{13}C_2$, all at 0.04 mM). Samples were sonicated, centrifuged and dried under nitrogen flow. Samples were reconstituted in 50 µL 99% MilliQ water with 1% MeOH and 0, 2% Formic acid.

Alternatively, urinary metabolites (human samples) were normalized by the concentration of creatinine, which was determined on diluted urines (1:25, with water) according to a published protocol (Behringer et al, 2019). Metabolites were quantified using a 12-point calibration curve created by serial dilution of solutions made with pure analytes dissolved in LC-MS quality water with the following concentration ranges: αKG: 0–500 µM, αKGM: 0–50 µM, and creatinine: 0–500 µM. $D_3$-methylmalonic acid (5 µM, for αKG and αKGM) and $D_3$-creatinine (25 µM, for creatinine) were used as internal standards. Calibrators, quality controls (urine sample of known creatinine concentration), and study plasma and urine samples were prepared by mixing 20 µL of sample with 20 µL freshly prepared dithiothreitol 0.5 M followed by vortexing and incubation at room temperature for 10 min. In total, 20 µL of internal standard mix was then added, and the samples were vortexed. Metabolites were extracted by the addition of 100 µL of 0.1% formic acid in methanol, followed by vortexing and centrifugation at 10,000 × g for 10 min at room temperature. The supernatants were transferred into HPLC vials, and 10 µL were injected for LC-MS/MS analysis.

### TCA-metabolite analysis

Samples (8 µL) were injected via an Infinity II Bio liquid chromatography system into a 6495C triple quadrupole mass spectrometer (both Agilent Technologies, Waldbronn/Germany). Metabolites were separated on an Acquity HSS T3 C18 column (1.8 µm, 2.1 × 150 mm, Waters) by using the following mobile phase binary solvent system and gradient at a flow rate of 0.35 mL/

min: Mobile A consisted of 100% water with 0.2% formic acid. Mobile phase B consisted of 100% acetonitrile with 0.2% formic acid. The following 11 min gradient program was used: 0 min 1% B, 0–6 min 1% B, 6–7 min 80% B, 7–8 min 80% B, 8–11 min 1% B. The column compartment was set to 30 °C. Metabolites were detected with authentic standards and/or via their accurate mass, fragmentation pattern, and retention time in polarity switching ionization dynamic MRM AJS-ESI mode, and quantified (where appropriate) via a calibration curve. The gas temperature of the mass spectrometer was set to 240 °C and the gas flow to 19 L/min. The nebulizer was set to 50 psi. The sheath gas flow was set to 11 L/min, with a temperature of 400 °C. The capillary voltages were set at 1000/1000 V with a nozzle voltage of 500/500 V. The voltages of the High-Pressure RF and Low-Pressure RF were set to 100/100 and 70/70 V, respectively. Metabolite peaks were annotated with Skyline-daily (version 22.2.1.278).

For human urine and plasma samples, αKG was determined as previously described (Moritz et al, 2023), and αKGM was compatible with the chromatographic conditions of the TCA panel. The two fragments of the parent ion ($m/z$ 105.1 Da and $m/z$ 91 Da) were monitored in positive mode with a Sciex 6500 + ESI-tripleQ MS/MS. Signal processing and quantification of metabolites were carried out with Analyst® 1.7.2 software, 2022 AB Sciex.

Metabolites are represented in concentrations when a standard curve was available or their signals were alternatively normalized to that of internal standards and expressed as intensity or peak area.

## Human samples

αKGM and αKG were measured in urine and serum that were collected from a total of 19 male and female individuals (average age of 46) recruited from the employees of the Saarland University Hospital. Only healthy subjects without prevalent cardiovascular or chronic kidney disease, diabetes, or regular medication were included. All participants gave their written consent, and the study was approved by the local institutional review committee (155/13).

## Animal procedure

Nit2-floxed and Nit2$^{KO}$ mice were generated by the Laboratory Animal Resource Center, University of Tsukuba, as described (Mizuno-Iijima et al, 2021). LoxP sites flanking exon 2 of *NIT2* were inserted by means of CRISPR/Cas9. Nit2-knockout mice (Nit2$^{ko/ko}$) were generated by a 597 bp deletion of exon 2 by CRISPR/Cas9. Endothelial cell-specific, tamoxifen-inducible knockout mice of NIT2 (ecNit2$^{-/-}$) were generated by crossing Nit2$^{flox/flox}$ mice (Nitrilase-like 2, Nit2$^{tm1NH}$) with Cdh5-CreERT2 (Tg(Cdh5-CreERT2)$^{1Rha}$) (Wang et al, 2010) mice (kindly provided by Ralf Adams, Münster, Germany). NIT2 deletion was induced by providing tamoxifen in the diet (400 mg/kg, 10 days) when male animals were at least 8 weeks old. A tamoxifen-free "wash out" period of at least 14 days after tamoxifen feeding was adhered to. Control animals (CTL) are defined as Nit2$^{flox/flox}$-Cdh5-CreERT2$^{0/0}$ littermates (i.e., no Cre expression) and were also treated with tamoxifen.

Systemic inflammation in mice (18 weeks old, C57Bl6/J, 29 ± 3 grams body weight) was induced by a single i.p. lipopolysaccharide injection (LPS from *Escherichia coli* strain O55:B5, 4 mg/kg, 4 h, Sigma-Aldrich, St. Louis, USA).

All animals had free access to chow and water in a specified pathogen-free facility with a 12 h day/ 12 h night cycle, and all animal experiments were performed in accordance with the German animal protection law and were carried out after approval by the local authorities (Regional council Darmstadt or Northrhine Westfalia, under the approval FU2020 or 81-02.04.2022.A334, respectively). Every mouse received an identification number for each experiment, and the experimenter was blind for the genotype or treatment. Animal group sizes differ due to the number of littermates. Control and knockout animals were studied in a paired fashion per experiment.

## Neonatal retina angiogenesis

One-day-old pups (ecNit2$^{-/-}$ and their control counterparts) were injected intragastrically with 50 μL of tamoxifen at 1 mg/mL dissolved in sunflower oil for 3 consecutive days (Pitulescu et al, 2010). On day 6, pups were injected with 5-ethynyl-2'-deoxyuridine (EdU, Click-iT Plus EdU Alexa Fluor 488 Imaging Kit from Invitrogen, C10637) four hours before being sacrificed. The eyes were removed and the retina was exposed and divided into four petals by incisions towards the its center. It was fixed with 100% methanol at −20 °C overnight. Retinas were washed and treated with 0.1% Triton X-100, 1% bovine serum albumin (BSA), and 1% donkey serum (Sigma, D9663). Next, the Click-iT reaction for EdU was performed, and retinas were stained with isolectin GS-IB4 (1:500, ThermoFisher, I21411) overnight at 4 °C. Retinas were mounted on microscope slides (Thermo Scientific-Superfrost, 631-9483) using Dako Fluorescence Mounting medium (Agilent Technologies Inc., S3023). Images were acquired with a Zeiss LSM800 laser scanning microscope (Carl Zeiss Microscopy GmbH) under a ×20 objective using the software ZEN (ZEN 3.1, Carl Zeiss Microscopy GmbH). Images were acquired using the tile mode (100 tiles per retina). Quantification of the vascular network was performed using the freely available software Angiotool64 (version 0.6a). EdU staining was quantified by counting its signal present only in vessels using Fiji ImageJ 1.54p.

## Choroidal neovascularization

To address the role of endothelial NIT2 for angiogenesis in vivo in the adult stage, we analyzed the choroidal neovascularization upon a laser-induced injury. Endothelial cells in this type of angiogenesis penetrate through Bruch's membrane into the normally avascular subretinal space. (Gong et al, 2015) Briefly, adult mice received tamoxifen to delete endothelial NIT2, and after the wash out time, four lesions were induced using an image-guided laser photocoagulation system (Phoenix MICRON IV Image-Guided Laser System, California, USA) under anethesia with Xylasine and Ketamine (6 and 100 mg/kg, respectively i.p.). Lesions were applied at double the disc diameter away from the optic nerve through an argon-laser pulse at 532 nm with a fixed diameter of 50 μm, a duration of 70 ms, and a power of 240 mW ± 60 mW. After 7 days, the mice were sacrificed, and the retinal pigment epithelium/choroid/sclera was flat-mounted, stained with isolectin B4 (1:500, ThermoFisher, I21411) and CD31 (1:200, R&D Systems, AF3628), and imaged. Quantification of isolectin-positive staining area was performed with Fiji ImageJ 1.54p and represented as mm².

## Aortic ring outgrowth assay

Aortic rings (1 mm) were embedded in a gel mixture of rat-tail collagen I (1.5 mg/mL, #354236, BD) 1× Medium 199 (#M0650, Sigma), and NaHCO₃ (2.2 mg/mL) at 37 °C for 60 min. EBM supplemented with 2.5% autologous serum was added to the gels. Aortic rings were treated or not with murine VEGF-165 (30 ng/mL, #450-32-10UG, PeproTech) and cultured for 7 days. Rings were fixed with 4% PFA, treated with 0.5% Triton X-100, and 1% BSA. Endothelial cells were stained with an anti-mouse CD31 antibody (1:200, #550274, BD). Images were acquired with a Zeiss LSM800 laser scanning microscope (Carl Zeiss Microscopy GmbH) Images were acquired using the tile- and Z- mode and sprouts were quantified using Angiotool64 (version 0.6a).

## Cell migration

A scratch wound assay was performed with HUVEC in endothelial growth media (EGM) in 96-well plates. Cell migration into the scratched area (Incucyte WoundMaker, wound closure) was monitored by live cell imaging and analyzed using the IncuCyte ZOOM platform.

## Proliferation assay

In total, 3000 HUVEC were seeded out onto a 96-well plate in endothelial growth media. Nuclei were stained using the IncuCyte Nuclight Rapid Red Dye according to the manufacturer's instructions. Proliferation was monitored by live cell imaging using the IncuCyte ZOOM platform.

## Spheroid outgrowth assay

Spheroid outgrowth assays were performed as previously described (Korff and Augustin, 1998). HUVEC spheroids were stimulated for 16 h with human recombinant VEGF-A (50 ng/mL). Images were generated with the Evos XL Core. The quantitative analysis of sprout number and cumulative length was calculated with the AxioVision software.

## RNA-sequencing

RNA and library preparation integrity were verified with LabChip Gx Touch 24 (Perkin Elmer). Sequencing was performed on NextSeq2000 instrument (Illumina) with 1 ×72 bp single end setup. The resulting raw reads were assessed for quality, adapter content, and duplication rates with FastQC (RRID:SCR_014583) (Simon, 2010).

Sequencing reads were aligned against the hg38 genome assembly using STAR (v2.7.10, RRID:SCR_004463), with the parameter – quantMode set to "GeneCounts". Differential gene expression analysis was performed using DESeq2 (v1.32.0; RRID:SCR_015687) in R (v4.1.1; R Project for Statistical Computing (RRID:SCR_001905) (Love et al, 2014; R Core Team (2021)). Differentially expressed genes were taken as those with a false discovery rate-adjusted *P* value of less than 0.05. Gene set enrichment analysis was performed using the gprofiler2 (v0.2.3, RRID:SCR_018190) package for R, using the WikiPathways (RRID:SCR_002134) source. F1 scores were calculated using the resulting precision and recall values, as standard. Visualizations of

the data and results were generated using the R package ggplot2 (RRID:SCR_014601) in R.

## β-Galactosidase assay

HUVEC were cultured in endothelial basal media (EBM) with 2% FCS for 16 h. Cells were fixed and treated following the manufacturer's instructions in the Senescence β-Galactosidase Staining Kit (#9860, Cell Signaling Technology). Senescence is expressed by the percentage of β-galactosidase-positive cells counted in at least three randomly chosen images per each of the three batches of HUVEC.

## Statistics

Data are represented as mean ± standard error of the mean. Calculations were performed with Prism 9.2.0. The latter was also used to test for normal distribution and similarity of variance. In the case of multiple testing, a Bonferroni correction was applied. For multiple group comparisons, analysis of variance followed by post hoc testing was performed. Individual statistics of dependent samples were performed by paired *t* test, of unpaired samples by unpaired *t* test, and, if not normally distributed, by the Mann–Whitney *U* test as indicated. *P* values of <0.05 were considered significant. Unless otherwise indicated, *n* indicates the number of individual experiments. Statistics for RNAseq, MS, and metabolomics were carried out as described in the specific sections.

# Data availability

Datasets provided as Excel files: Excel Dataset 1: Differential gene expression analysis (RNAseq) of HUVEC NTC, NIT2⁻/⁻, GLS1⁻/⁻ and NIT2/GLS1⁻/⁻. Excel Dataset 2: Phenome-wide association studies to rs38380303. Excel Dataset 3: Phenome-wide association studies to rs277627. Excel Dataset 4: Untargeted metabolomics of HUVEC NTC, NIT2⁻/⁻, GLS1⁻/⁻ and NIT2/GLS1⁻/⁻. Excel Dataset 5: Untargeted metabolomics of plasma from CTL and ecNit2−/− mice. Excel Dataset 6: Untargeted metabolomics of lung tissue from CTL and ecNit2⁻/⁻ mice. The datasets produced in this study are available in the following databases: (1) The mass spectrometry proteomics data have been deposited to the ProteomeXchange Consortium via the PRIDE (Perez-Riverol et al, 2022) partner repository with the dataset identifier PXD055885. (2) RNA-seq data: RNA-seq data of HUVEC NTC, NIT2−/− GLS1−/− and NIT2/GLS1−/− samples, along with HUVEC treated with aKGM or control for 4 h or 24 h are available at GEO under the accession GSE302822.

The source data of this paper are collected in the following database record: biostudies:S-SCDT-10_1038-S44318-025-00642-7.

# Peer review information

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

## Acknowledgements

We dedicate this study to Prof. Arthur JL Cooper (* 04.19.1946 † 05.30.2024), who was an emeritus professor at Medical College in Valhalla, NY, and adjunct professor at Weill Cornell Medical College, NY, United States. He studied NIT2 and the GTωA pathway in the 1970s and had a substantial intellectual contribution to the present study. We are grateful for the excellent technical assistance of Susanne Wienströer, Zahraa Jaber, Agnes Krüger, Katalin Pálfi, Jana Meisterknecht, Manuela Späth, and Natalie Weber. The study was supported by the Deutsche Forschungsgemeinschaft (DFG): SFB 1531 –Project number 456687919 to RPB, FR, IF, MS, SD, ISB, AAP, MHS and IW. Germany's Excellence Strategy CIBSS – EXC-2189 – Project ID 390939984 to LH. Excellent Cluster Cardio-Pulmonary Institute (EXS2026, project number 390649896), the Medicine Faculty of the Goethe University (University of Frankfurt, Germany); the German Center for Cardiovascular Research (DZHK), Partner Site Rhein-Main, Frankfurt am Main.

## Author contributions

**Niklas Herrle**: Conceptualization; Data curation; Investigation; Methodology. **Pedro F Malacarne**: Data curation; Formal analysis; Investigation; Visualization; Methodology. **Timothy Warwick**: Data curation; Formal analysis. **Alfredo Cabrera-Orefice**: Data curation; Formal analysis; Visualization; Methodology. **Yiheng Chen**: Data curation; Formal analysis; Investigation; Methodology. **Maedeh Gheisari**: Data curation; Formal analysis; Investigation. **Souradeep Chatterjee**: Data curation; Formal analysis; Investigation; Methodology. **Matthias S Leisegang**: Conceptualization; Data curation; Formal analysis; Investigation; Methodology. **Tamim Sarakpi**: Resources. **Sarah Wionski**: Investigation. **Melina Lopez**: Investigation. **Carine Kader**: Investigation. **Tom Teichmann**: Investigation. **Maria-Kyriaki Drekolia**: Investigation. **Ina Koch**: Investigation. **Marcus Keßler**: Formal analysis. **Sabine Klein**: Resources; Investigation. **Frank Erhard Uschner**: Resources. **Jonel Trebicka**: Resources. **Steffen Brunst**: Investigation. **Ewgenij Proschak**: Investigation. **Stefan Günther**: Formal analysis; Investigation. **Mónica Rosas-Lemus**: Data curation; Investigation; Visualization; Methodology. **Nina Baumgarten**: Data curation; Formal analysis; Methodology. **Stephan Klatt**: Data curation; Formal analysis; Investigation; Methodology. **Thimoteus Speer**: Resources. **Sofia-Iris Bibli**: Resources. **Marta Segarra**: Investigation; Methodology. **Amparo Acker-Palmer**: Resources. **Julian U G Wagner**: Investigation. **Ilka Wittig**: Data curation; Formal analysis; Investigation; Methodology. **Stefanie Dimmeler**: Resources; Funding acquisition. **Marcel H Schulz**: Data curation; Formal analysis; Funding acquisition; Methodology. **JB Richards**: Resources. **Ralf Gilsbach**: Formal analysis; Methodology. **Travis T Denton**: Conceptualization. **Ingrid Fleming**: Resources; Funding acquisition. **Luciana Hannibal**: Conceptualization; Data curation; Formal analysis; Funding acquisition; Validation; Investigation. **Ralf P Brandes**: Conceptualization; Resources; Writing—original draft; Writing—review and editing. **Flávia Rezende**: Conceptualization; Data curation; Formal analysis; Supervision; Funding acquisition; Investigation; Methodology; Writing—original draft; Project administration; Writing—review and editing.

Source data underlying figure panels in this paper may have individual authorship assigned. Where available, figure panel/source data authorship is listed in the following database record: biostudies:S-SCDT-10_1038-S44318-025-00642-7.

## Funding

## Disclosure and competing interests statement

The authors declare no conflict of interest that could potentially influence or bias the work. TD co-owns LiT Biosciences. The company did not contribute funding to the current manuscript or to any work previously carried out by the authors. JBR is the CEO of 5 Prime Sciences (www.5primesciences.com), which provides research services for biotech, pharma, and venture capital companies for projects unrelated to this research. He has served as an advisor to

GlaxoSmithKline and Deerfield Capital. JBR's institution has received investigator-initiated grant funding from Eli Lilly, GlaxoSmithKline, and Biogen for projects unrelated to this research. YC is an employee of 5 Prime Sciences. JT has received speaking and/or consulting fees from Astra-Zeneca, Gore, Boehringer-Ingelheim, Falk, Grifols, Genfit, and CSL Behring.

# Expanded View Figures

**A**

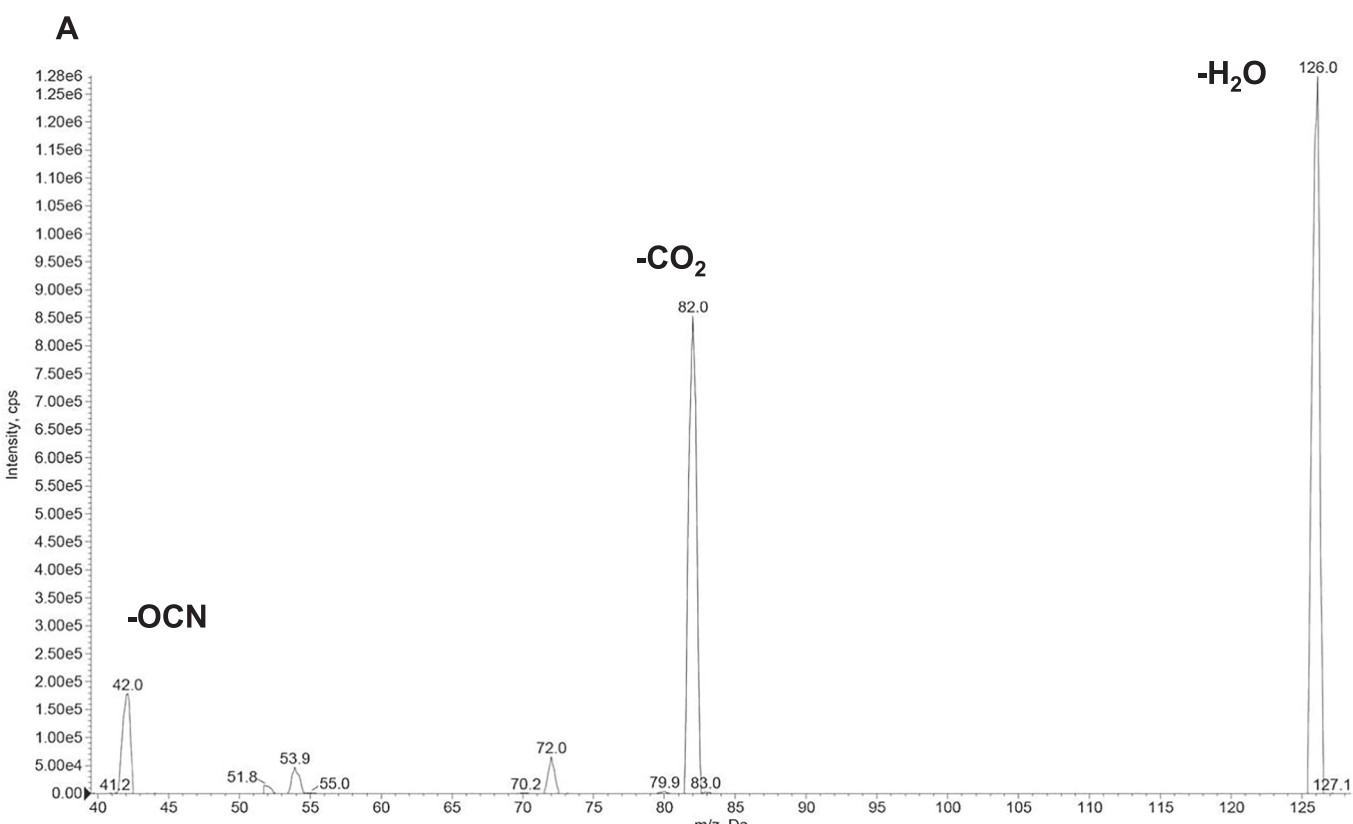

**B**

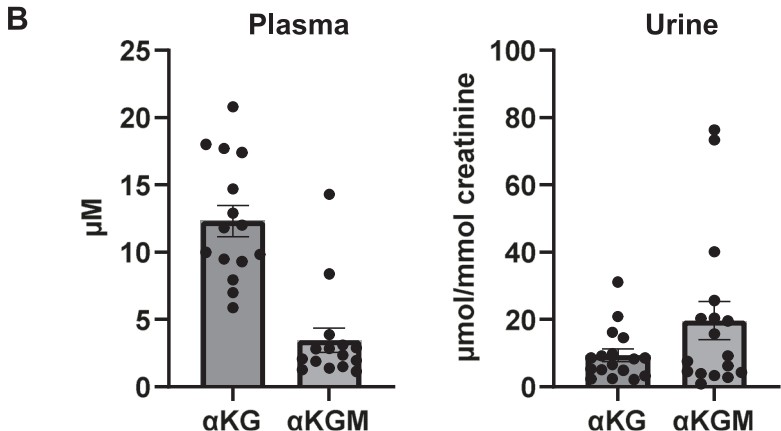

Figure EV1.   Targeted LC-MS/MS of αKGM.

(A) The mass spectrum of αKGM in its lactam form as detected in negative ionization mode. The spectrum shows three major fragmentation peaks at 126.0 Da (M-1-H2O; -18 Da), 82.0 Da (M-1-CO2-H2O; -62 Da) and 42.0 Da (a CNO- fragment). (B) αKG and αKGM concentrations (using calibration curves of known concentrations of each compound) in plasma and urine (normalized to creatinine) of healthy individuals. Source data are available online for this figure.

**A**

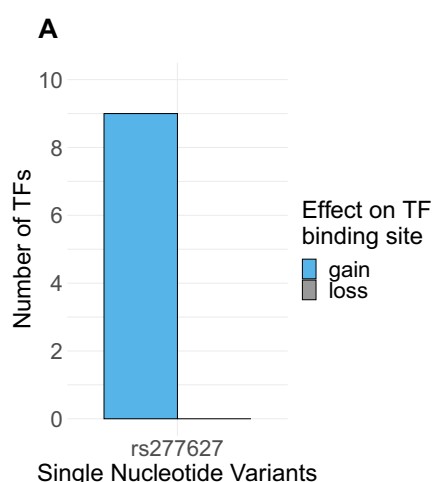

Number of TFs

Effect on TF binding site
- gain
- loss

rs277627

Single Nucleotide Variants

**B**

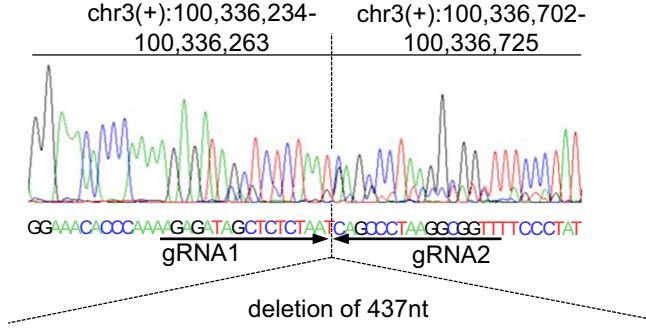

chr3(+):100,336,234-100,336,263 | chr3(+):100,336,702-100,336,725

GGAAACACCCAAAAGAGATAGCTCTCTAATCAGCCCTAACGGCGGTTTTCCCTAT

gRNA1    gRNA2

deletion of 437nt

within NIT2 Intron1 chr3(+)(hg38):100,336,264-100,336,701
incl. SNV at chr3:100,336,429

>deleted_region_hg38_dna range=chr3:100336264-100336701
GGTAGGATTTTCAGACTCTGGAGCAGAGGTGGGATGGGAAGTTCGGGCAGGCCATTAC
TAATTGAAGGGGTGGGTTGCCCCTCCACACCTGTGGGTGTTTCTCGTTAGGTGGAACG
AGAGACTTGGAAAAGAAAAAGACACAAACAAAGTATAGAGAAAGAAGTAGGGGGGCCC
AGGGGACCAGCGTTCGGCATACGGAGGATCCCGTCGGCCTCTGAGTTCCCTTAGTATT
TATGATCATTCTTGGGTGTTTCTCGGAGAGGGGGATGTGGCAGGGTCATAGGATAATAG
TGGAGAGAAGGTCAGCAGATAAACACGTGAACAAAGGTCTCTGCATCATGAACAAGGTA
AAGAATTAAGTGCTGTGCTTTAGATATGCATACACATAAACATCTCAATGTCTTAAAGAGC
AGTATTGCTGCCCGCCTGTCCCCCTTC

**C**

**NIT2 rs277627**

Aorta    Tibial artery

Norm. Expression

GG (37)   GA (166)   AA (184)

pvalue: 6.88e-9

Norm. Expression

GG (37)   GA (166)   AA (184)

pvalue: 6.88e-9

**D**

NCBI Transcript Accession    NM_020202.4

Exon | Intron | guide RNA | Reference Sequence | SNP/MNP | Insertion | Deletion

100336396    100336461

5'    3'

A
T

AAAGACACAAACAAAGTATAGAGAAAGAAGTA G GGGGCCCAGGGGACCAGCGTTCGGCATACGG
TTTCTGTGTTTGTTTCATATCTCTTTCTTCAT C CCCCGGGTCCCCTGGTCGCAAGCCGTATGCC

**E**

NIT2 G-A    gttaggtggaacgagagacttggaaaagaaaaagacacaaacaaagtatagagaaagaagtaAggggggcccaggggaccagcgttcggcatacggaggatcccgtcggcctctgagttcccttagtatttatga
NIT2 WT    gttaggtggaacgagagacttggaaaagaaaaagacacaaacaaagtatagagaaagaagtaGggggggcccaggggaccagcgttcggcatacggaggatcccgtcggcctctgagttcccttagtatttatga
complement

NIT2 G-A

AGACTTGGAAAAGAAAAAGACACAAACAAAGTATAGAGAAAGAAGTAAGGGGGCCCAGGGGACCAGCGTTCGGCATACGGAGGATCCCGTCGI
complement

**Sanger Seq**

NIT2 WT

GACTTGGAAAAGAAAAAGACACAAACAAAGTATAGAGAAAGAAGTAGGGGGGCCCAGGGGACCAGCGTTCGGCATACGGAGGATCCCGTCGI

◀ **Figure EV2. Regulation of *NIT2* expression by the SNV rs277627.**

(A) rs277627 is located in a regulatory element (REM, analyzed with EpiRegioDB) with binding sites for the transcription factors ZNF320 VEZF1 KLF16 GLIS2 MAZ ZNF740 ZNF148 ZNF467 GLIS3 that are expressed in HUVEC and have a gain of function and likely repress transcription. (B) A two gRNA approach was employed to delete a 437 bp region where the SNV rs277627 (chr3:100336428–100336429) in NIT2 (intron 1) is located. gRNA1 targets the chr3:100,336,234-100,336,263 region whereas gRNA2 the chr3:100,336,702-100,336,725. A successful deletion was obtained between 100,336,264-100,336,701 as shown by Sanger sequencing. The SNV G is labeled in red. (C) Data from GTex for the expression of wild-type (GG) rs277627 and its variants GA and AA on NIT2 expression in human aorta and tibial artery. (D) Strategy of the CRISPR/Cas9-mediated mutation to insert the NIT2 intron containing relevant SNP rs277627. Genomic locus of the human NIT2 (NM_020202.4) intron and mutation of G (WT) to an A (rs277627). (E) Sanger Sequencing of genomic DNA after CRISPR/Cas9-mediated generation of the rs277627.

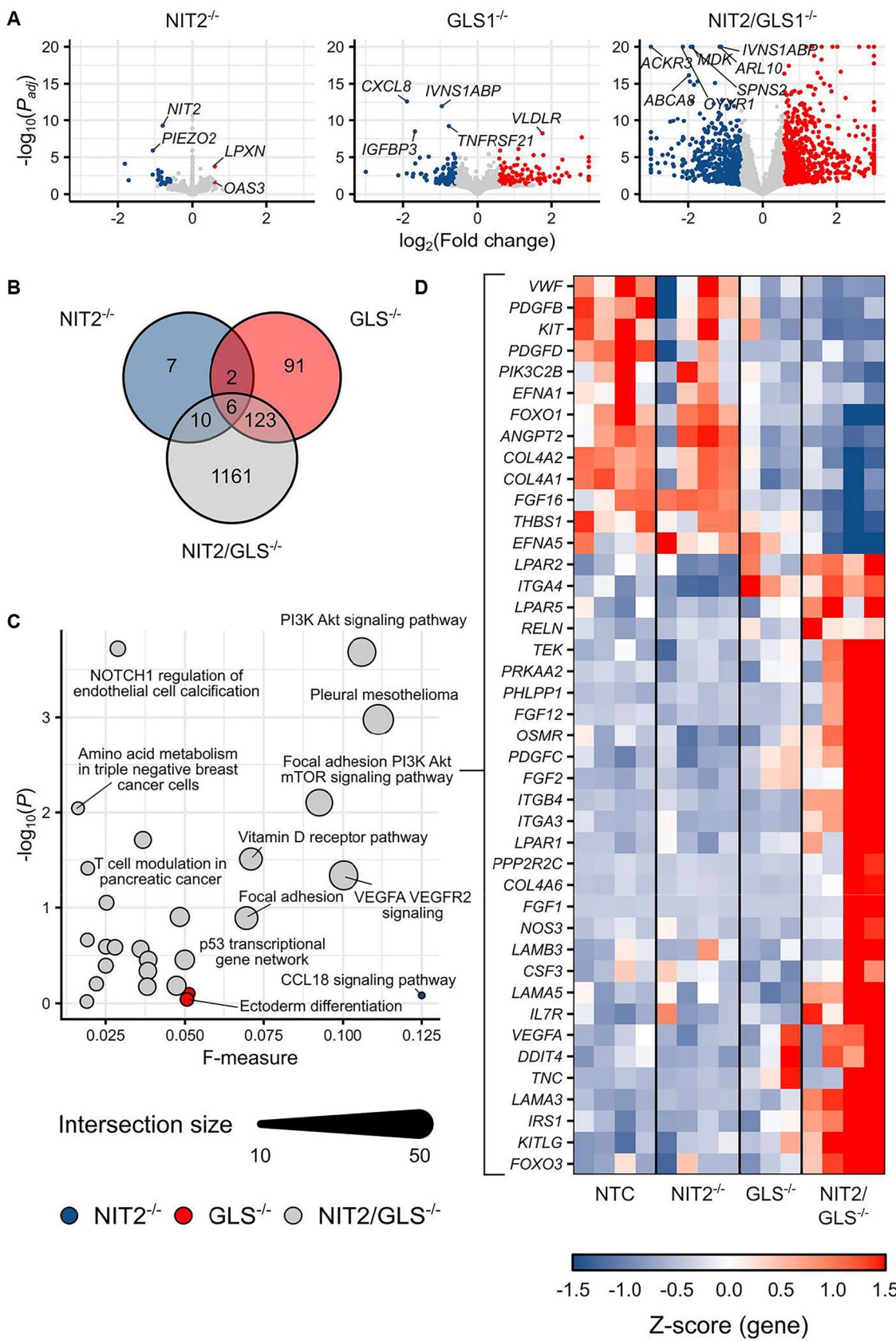

**Figure EV3.   NIT2 and GLS1 sinergistically control gene expression (RNAseq) in HUVEC.**

(**A**) Volcano plots for each knockout as indicated. (**B**) Number of differentially expressed genes. (**C**) Annotation pathway. (**D**) Heatmap of top differentially expressed genes in HUVEC NTC, NIT2$^{-/-}$, GLS1$^{-/-}$ and NIT2/GLS1$^{-/-}$.

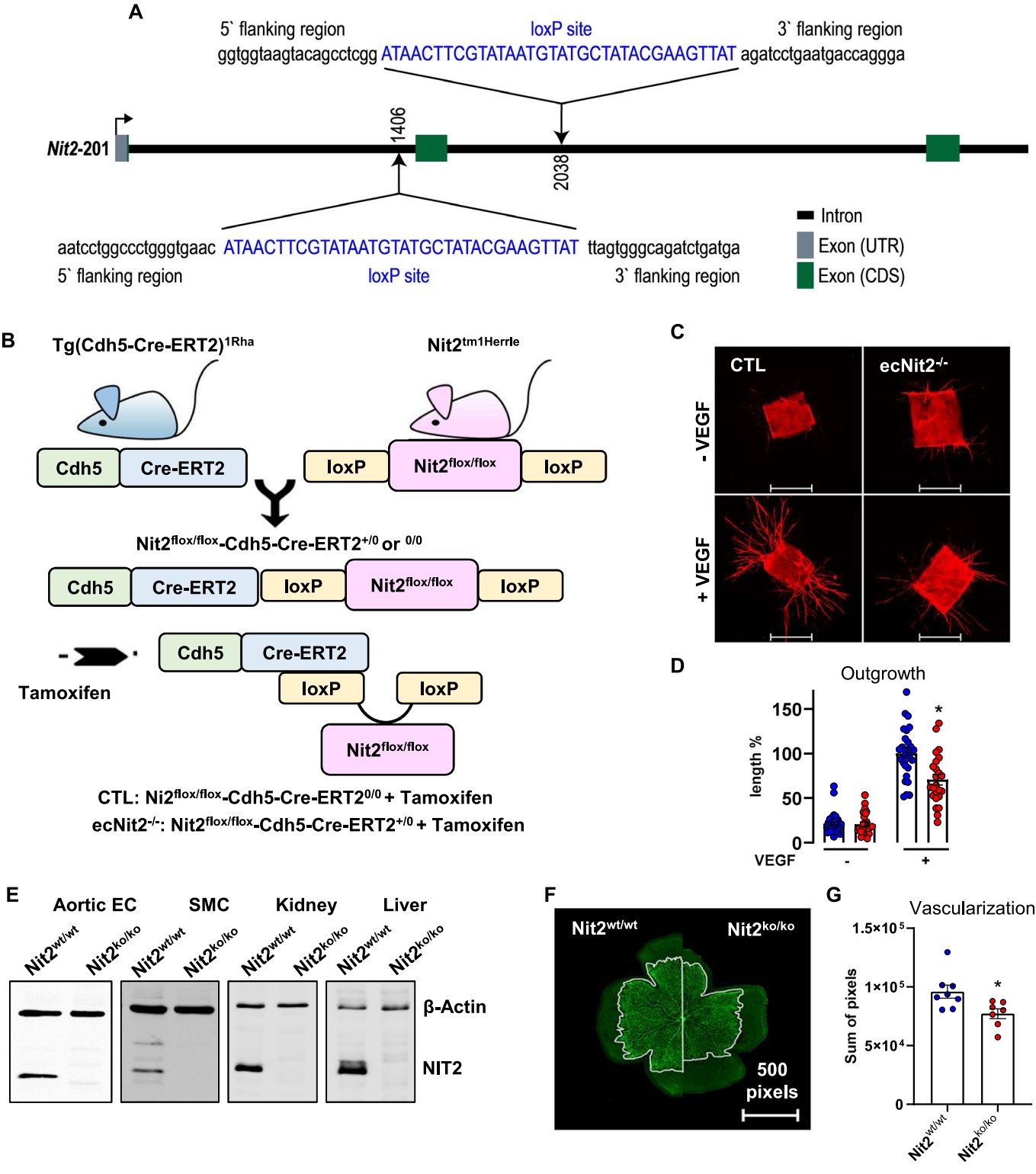

**A**

5` flanking region
ggtggtaagtacagcctcgg

loxP site
ATAACTTCGTATAATGTATGCTATACGAAGTTAT

3` flanking region
agatcctgaatgaccaggga

Nit2-201

1406

2038

aatcctggccctgggtgaac
5` flanking region

ATAACTTCGTATAATGTATGCTATACGAAGTTAT
loxP site

ttagtgggcagatctgatga
3` flanking region

■ Intron
■ Exon (UTR)
■ Exon (CDS)

**B**

Tg(Cdh5-Cre-ERT2)$^{1Rha}$    Nit2$^{tm1Herrle}$

Cdh5 | Cre-ERT2    loxP | Nit2$^{flox/flox}$ | loxP

Nit2$^{flox/flox}$-Cdh5-Cre-ERT2$^{+/0}$ or $^{0/0}$

Cdh5 | Cre-ERT2 | loxP | Nit2$^{flox/flox}$ | loxP

Tamoxifen

Cdh5 | Cre-ERT2
loxP | loxP
Nit2$^{flox/flox}$

CTL: Ni2$^{flox/flox}$-Cdh5-Cre-ERT2$^{0/0}$ + Tamoxifen
ecNit2$^{-/-}$: Nit2$^{flox/flox}$-Cdh5-Cre-ERT2$^{+/0}$ + Tamoxifen

**C**

CTL    ecNit2$^{-/-}$

− VEGF

+ VEGF

**D**

Outgrowth

length %

VEGF   −   +

*

**E**

Aortic EC    SMC    Kidney    Liver

Nit2$^{wt/wt}$  Nit2$^{ko/ko}$   Nit2$^{wt/wt}$  Nit2$^{ko/ko}$   Nit2$^{wt/wt}$  Nit2$^{ko/ko}$   Nit2$^{wt/wt}$  Nit2$^{ko/ko}$

β-Actin

NIT2

**F**

Nit2$^{wt/wt}$    Nit2$^{ko/ko}$

500 pixels

**G**

Vascularization

Sum of pixels

Nit2$^{wt/wt}$    Nit2$^{ko/ko}$

*

◀ **Figure EV4. Generation of Nit2 knockout mice.**

(A) LoxP sites flanking exon 2 of the *NIT2* gene were inserted by CRISPR/cas9. (B) Nit2$^{flox/flox}$ mice were crossed with Tg(Cdh5-Cre-ERT2)$^{1Rha}$ to generate endothelial-specific, tamoxifen-inducible knockout mice of Nit2. (C) Ex vivo endothelial cell outsprouting from aortic segments isolated from CTL and ecNit2$^{-/-}$ mice with quantification, normalized to CTL −VEGF (D). *$P < 0.05$, Mann–Whitney test (-VEGF versus +VEGF). (E) Validation of NIT2 deletion by Western blot in aortic endothelial cells as well as other tissue of a global, constitutive knockout mouse of Nit2 (Nit2$^{wt/wt}$ and Nit2$^{ko/ko}$, generated by CRISPR/cas9 deletion of ~500 bp. Nit2$^{wt/wt}$ and Nit2$^{ko/ko}$ are littermates in a heterozygous breeding of Nit2$^{wt/ko}$ with Nit2$^{wt/ko}$ mice.). (F) Retina angiogenesis in neonatal mice with quantification (G). *$P < 0.05$ Nit2$^{wt/wt}$ as compared to Nit2$^{ko/ko}$, Mann–Whitney test. SMC aortic smooth muscle cells. Source data are available online for this figure.

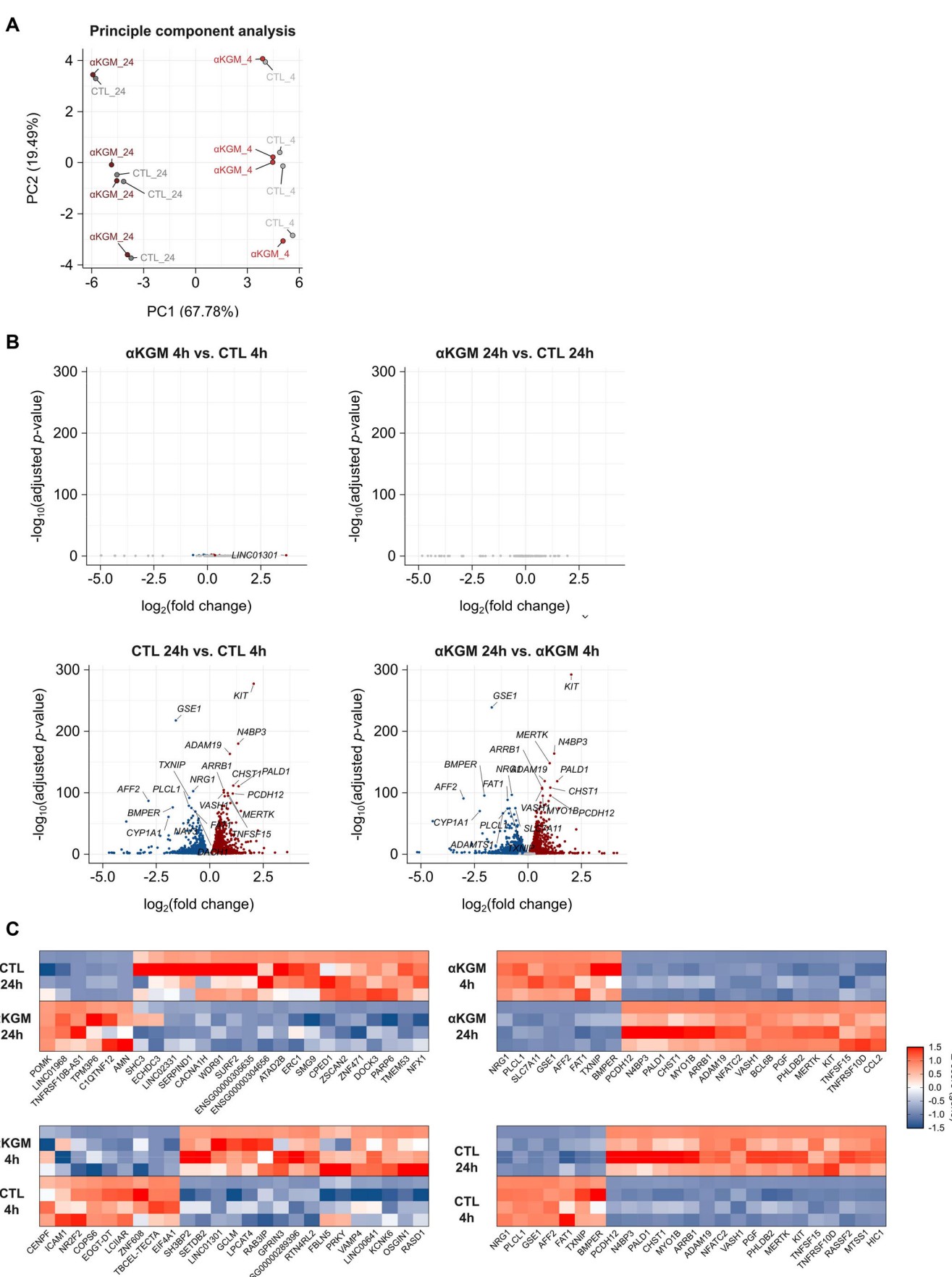

**Figure EV5. Effect of αKGM on gene expression in endothelial cells.**

(**A**) PCA analysis, volcano plots (**B**) and heat maps (**C**) of gene expression (RNAseq) in HUVEC without (CTL) or with 2-hydroxy-5-oxo-proline (300 μM, αKGM) after 4 or 24 h of exposure to the compound. In this analysis, only time segregates the sample groups but not the treatment with αKGM suggesting that the compound is inert once it is secreted by cells.

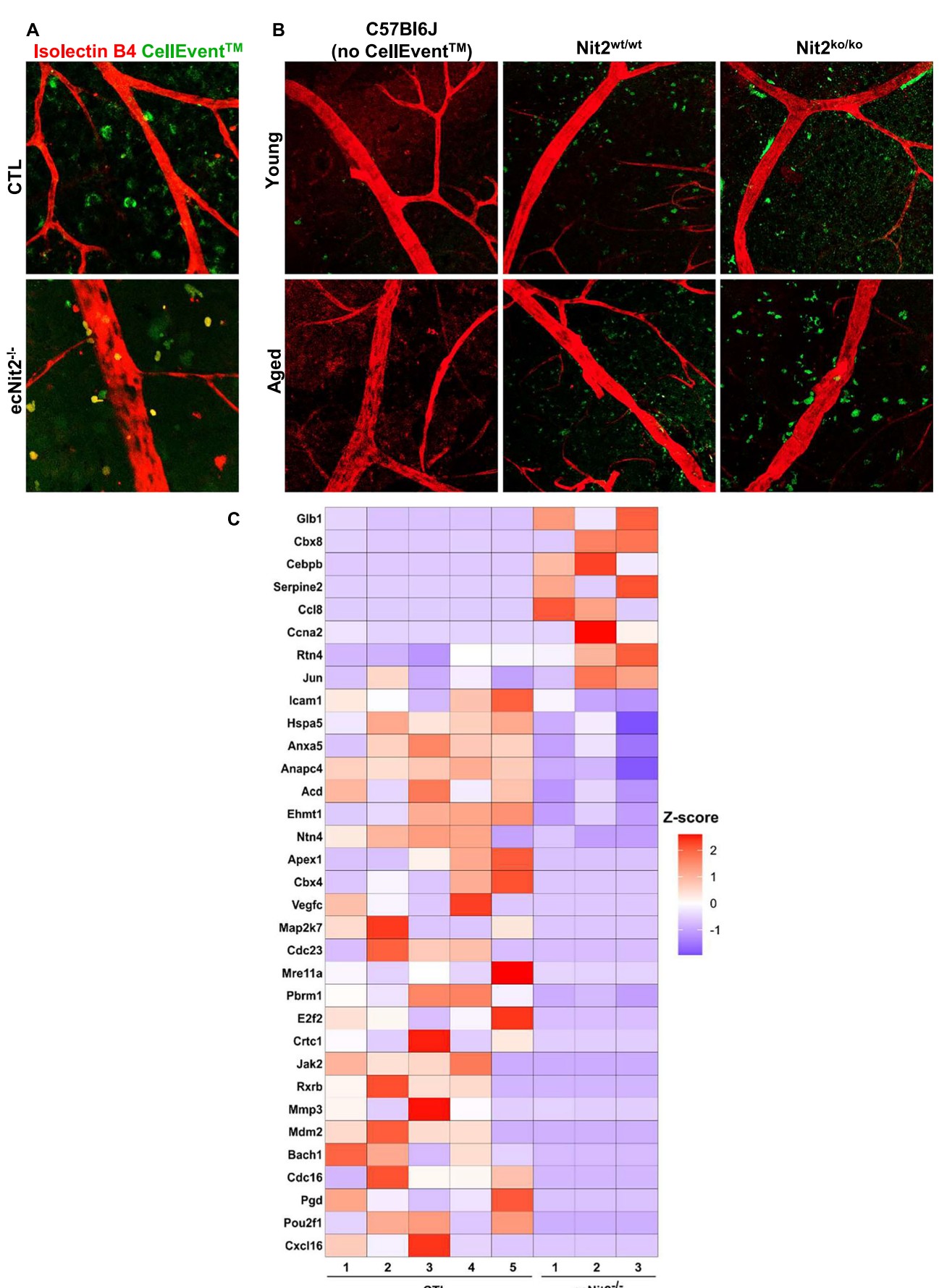

◀  **Figure EV6.  Cellular senescence in knockout mice of NIT2.**

CellEvent™ and isolectin B4 staining in retinae of CTL and ecNit2$^{-/-}$ mice (**A**) or Nit2$^{wt/wt}$ and Nit2$^{ko/ko}$ (**B**) young (3 months) and aged (9 months). (**C**) Differentially and significantly expressed genes (RNAseq with MACE) annotated to senescence in endothelial cells enriched from carotid arteries of CTL and ecNit2$^{-/-}$ mice. Source data are available online for this figure.

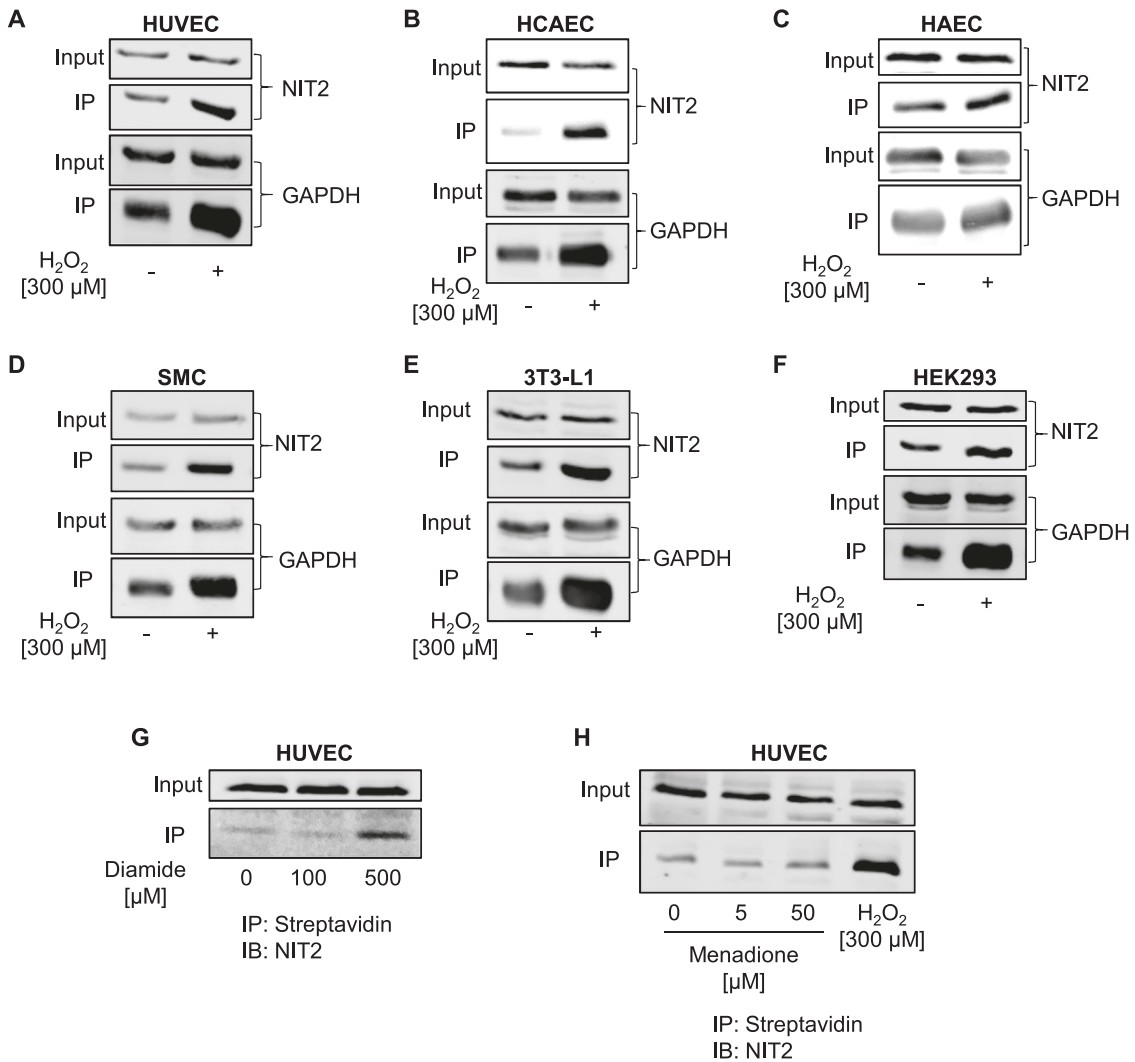

**Figure EV7. Cysteine oxidation in NIT2.**

Biotinylated iodoacetamide (BIAM) switch assay in different cell types exposed or not to 300 μM $H_2O_2$, 15 min. (A) HUVEC, (B) HCAEC, (C) HAEC, (D) SMC, (E) mouse fibroblasts 3T3-L1, (F) HEK 293. BIAM switch assay in HUVEC exposed to diamide (G) or menadione (H). HUVEC human umbilical vein endothelial cells, HCAEC human coronary artery endothelial cells, HAEC human aortic endothelial cells, SMC smooth muscle cells, IP immunoprecipitation, IB immunoblotting, GAPDH Glyceraldehyde-3-Phosphate Dehydrogenase. Source data are available online for this figure.

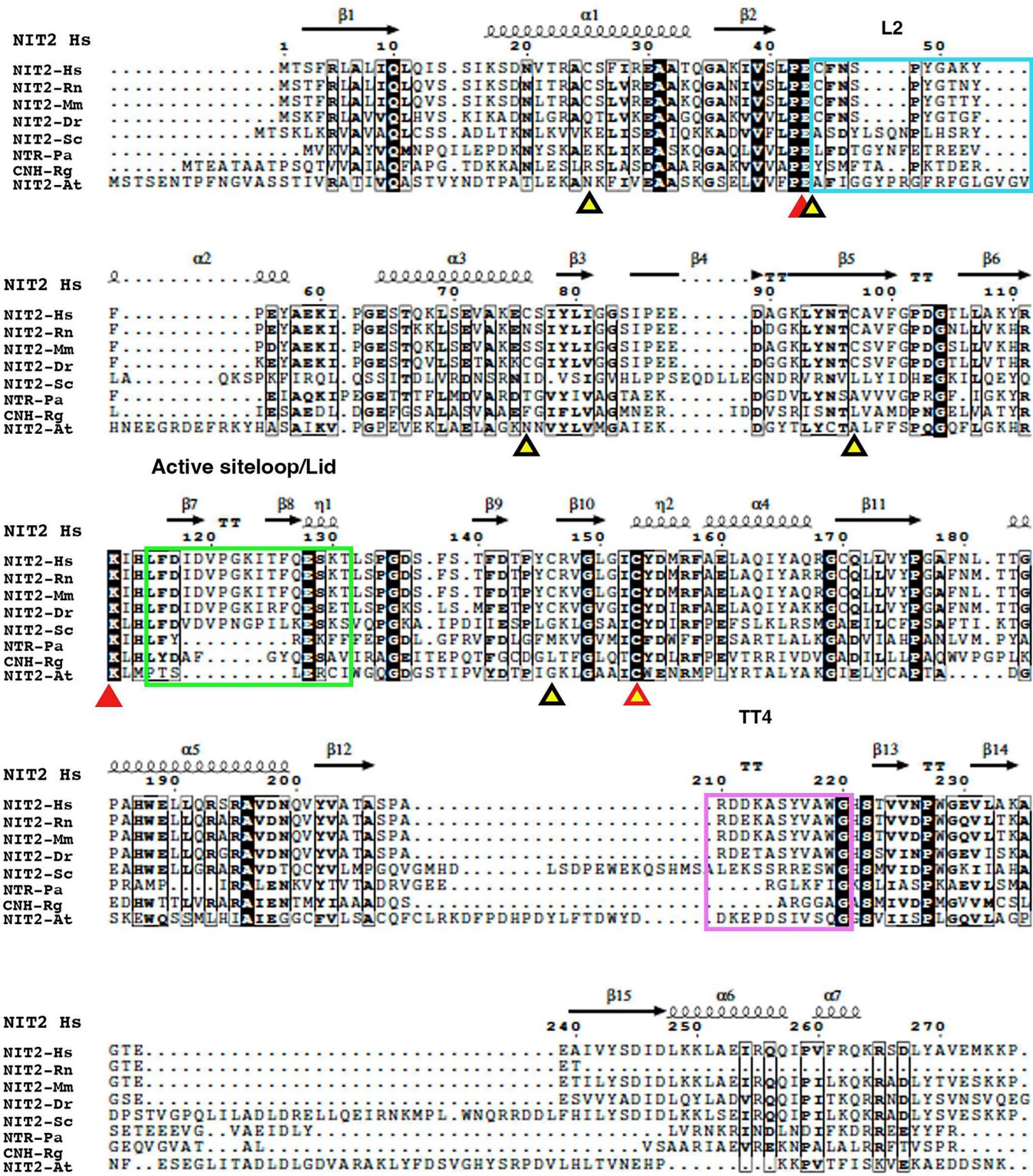

**Figure EV8. Animal NIT2 contains more cysteine residues than that of other organisms or plants.**

Multiple sequence alignment and structural features of human NIT2 predicted by AlphaFold2. Abreviations: Hs, *Homo sapiens* (Q9NQR4); Rn, *Rattus norvegicus* (Q497B0); Mm, *Mus musculus* (Q9JHW2); Dr, *Danio rerio* (Q4VBV9); Sc, *Saccharomyces cerevisiae* (P47016); Pa, *Pyrococcus abyssi* (Q9UYV8); Rg, *Rhodococcus qingshengii* (A0AA46MND0); At, *Arabidopsis thaliana* (P32962). The human cysteine residues are indicated with yellow triangles, the catalytic residues as red triangles, and the loops involved in the formation of the substrate channel and active site are colored in blue, green, and pink squares.

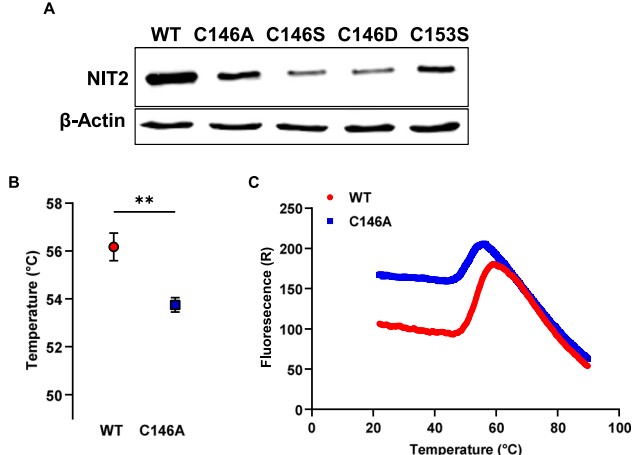

**Figure EV9. Expression and stability of NIT2 C146 mutants.**

(A) Western blot analysis (using an anti-His-tag antibody) for the His-tagged mutants of NIT2 C146 expressed in HEK 293 cells. (B) Aggregation points of NIT2 and NIT2 C146A as determined by thermal shift assay. $n = 3$; **$P < 0.01$, Welchs' correction. (C) Melting curve of purified NIT2 and NIT2 C146A protein. Source data are available online for this figure.

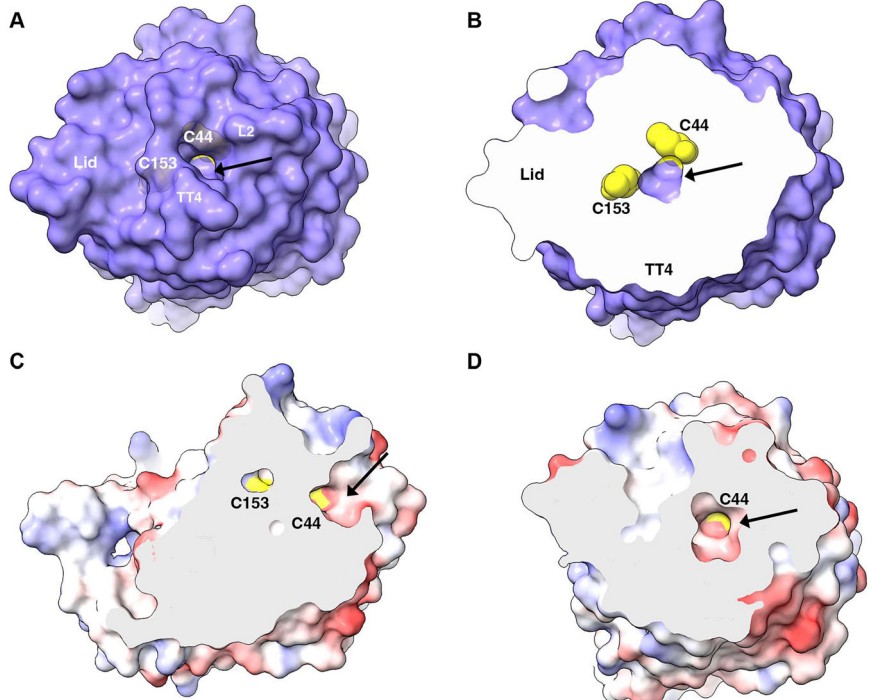

**Figure EV10. Cysteine 44 forms the substrate channel and is prone to oxidation.**

The AlfaFold2 structure prediction of NIT2 is depicted as the solvent-exposed surface in purple, and the substrate channel entry is shown from the top view (**A**) and the cut view to the bottom (**B**). The surface charge representation (red negative, white hydrophobic and blue positive) is shown on the side cut view (**C**) and top cut (**D**). Cys153 and Cys44 depicted as yellow spheres. The black arrow indicates the channel entry.

