## [Peer Review File · The EMBO Journal]

The transaminase- ω -amidase pathway senses oxidative stress to control glutamine metabolism and α -ketoglutarate levels in endothelial cells

Niklas Herrle, Pedro F. Malacarne, Timothy Warwick, Alfredo Cabrera-Orefice, Yiheng Chen, Maedeh Gheisari, Souradeep Chatterjee, Matthias Leisegang, Tamim Sarakpi, Sarah Wionski, Melina Lopez, Carine Kader, Tom Teichmann, Maria-Kyriaki Drekolia, Ina Koch, Mónica Rosas-Lemus, Marcus KeÄler, Sabine Klein, Frank Erhard Uschner, Jonel Trebicka, Steffen Brunst, Ewgenij Proschak, Stefan Günther, Nina Baumgarten, Stephan Klatt, Thimoteus Speer, Sofia-Iris Bibli, Marta Segarra, Amparo Acker-Palmer, Julian Wagner, Ilka Wittig, Stefanie Dimmeler, Marcel Schulz, J. Brent Richards, Ralf Gilsbach, Travis T. Denton, Ingrid Fleming, Luciana Hannibal, Ralf Brandes, and Flavia Rezende

Corresponding authors: Flavia Rezende (rezende@vrc.uni-frankfurt.de) , Ralf Brandes (brandes@vrc.uni-frankfurt.de)

Review Timeline:

Submission Date:	11th Feb 25
Editorial Decision:	25th Mar 25
Revision Received:	4th Jul 25
Editorial Decision:	12th Sep 25
Revision Received:	26th Sep 25
Accepted:	15th Oct 25

Editor: Daniel Klimmeck

Transaction Report:

Dear Dr Rezende,

Thank you again for the submission of your manuscript (EMBOJ-2025-120471) to The EMBO Journal, and providing us with a preliminary point-by-point response to the concerns raised by the referees. As mentioned, your study was assessed by two reviewers with expertise in cellular and systemic metabolism as well as vascular biology, whose comments are enclosed below.

As you will see from their comments, the referees acknowledge the potential interest and value of your findings. However, they also express important issues regarding the completeness of your study of the results, which need to be addressed thoroughly to make them supportive of publication in the EMBO Journal. In more detail, referee #1 points to a lack of sufficient support for physiological relevance of the proposed NIT2-aKGM-metabolic axis in endothelial cells, and overall quality and resolution of the related *in vivo* data presented (ref#1, standfirst, pt.1.; see also ref#2, pts.2,4). The reviewers also state that various claims and aspects related to the roles of adenine, senescence induction and the stable degradation product of aKGM, lactam, are underdeveloped currently, which dampens their enthusiasm for the study (ref#1, pts. 2,3; ref#2, pt.1). Further, the reviewers raise a number of issues related to the presentation of the findings, additional controls and improved methods annotation required, statistics applied and overall discussion of related literature, that would need to be conclusively addressed to achieve the level of robustness and clarity needed for The EMBO Journal.

Given the overall interest stated and broader angle of your results, we are able to invite you to revise your manuscript experimentally to address the referees' comments, along the lines sketched in your outline. I need to stress though that we do require strong support from the referees on a revised version of the study in order to move on to publication of the work.

Please feel free to contact me if you have any questions or need further input on the referee comments.

When submitting your revised manuscript, please carefully review the instructions below.

Please feel free to approach me any time should you have additional questions related to this.

Thank you for the opportunity to consider your work for publication.

I look forward to your revision.

Kind regards,

Daniel Klimmeck

Daniel Klimmeck, PhD
Senior Editor
The EMBO Journal

Instruction for the preparation of your revised manuscript:

2) individual production quality figure files as .eps, .tif, .jpg (one file per figure).

3) a .docx formatted letter INCLUDING the reviewers' reports and your detailed point-by-point response to their comments. As part of the EMBO Press transparent editorial process, the point-by-point response is part of the Review Process File (RPF), which will be published alongside your paper.

4) a complete author checklist, which you can download from our author guidelines ([https://wol-prod-cdn.literatumonline.com/pb-assets/embo-site/Author Checklist%20-%20EMBO%20J-1561436015657.xlsx](https://wol-prod-cdn.literatumonline.com/pb-assets/embo-site/Author%20Checklist%20-%20EMBO%20J-1561436015657.xlsx)). Please insert information in the checklist that is also reflected in the manuscript. The completed author checklist will also be part of the RPF.

6) It is mandatory to include a 'Data Availability' section after the Materials and Methods. Before submitting your revision, primary datasets produced in this study need to be deposited in an appropriate public database, and the accession numbers and database listed under 'Data Availability'. Please remember to provide a reviewer password if the datasets are not yet public (see <https://www.embopress.org/page/journal/14602075/authorguide#datadeposition>).

7) Our journal encourages inclusion of *data citations in the reference list* to directly cite datasets that were re-used and obtained from public databases. Data citations in the article text are distinct from normal bibliographical citations and should directly link to the database records from which the data can be accessed. In the main text, data citations are formatted as follows: "Data ref: Smith et al, 2001" or "Data ref: NCBI Sequence Read Archive PRJNA342805, 2017". In the Reference list, data citations must be labeled with "[DATASET]". A data reference must provide the database name, accession number/identifiers and a resolvable link to the landing page from which the data can be accessed at the end of the reference. Further instructions are available at .

8) At EMBO Press we ask authors to provide source data for the main and EV figures. Our source data coordinator will contact you to discuss which figure panels we would need source data for and will also provide you with helpful tips on how to upload and organize the files.

Numerical data can be provided as individual .xls or .csv files (including a tab describing the data). For 'blots' or microscopy, uncropped images should be submitted (using a zip archive or a single pdf per main figure if multiple images need to be supplied for one panel). Additional information on source data and instruction on how to label the files are available at .

9) We replaced Supplementary Information with Expanded View (EV) Figures and Tables that are collapsible/expandable online (see examples in <https://www.embopress.org/doi/10.15252/emboj.201695874>). A maximum of 5 EV Figures can be typeset. EV Figures should be cited as 'Figure EV1, Figure EV2' etc. in the text and their respective legends should be included in the main text after the legends of regular figures.

11) For data quantification: please specify the name of the statistical test used to generate error bars and P values, the number (n) of independent experiments (specify technical or biological replicates) underlying each data point and the test used to calculate p-values in each figure legend. The figure legends should contain a basic description of n, P and the test applied. Graphs must include a description of the bars and the error bars (s.d., s.e.m.).

Please remember: Digital image enhancement is acceptable practice, as long as it accurately represents the original data and conforms to community standards. If a figure has been subjected to significant electronic manipulation, this must be noted in the

figure legend or in the 'Materials and Methods' section. The editors reserve the right to request original versions of figures and the original images that were used to assemble the figure.

The revision must be submitted online within 90 days; please click on the link below to submit the revision online before 23rd Jun 2025.

Referee #1:

Summary:

Herrle and colleagues examined the metabolic response of endothelial cells to oxidative stress, with a focus on identifying redox-sensitive metabolic enzymes. Their findings reveal that reactive oxygen species (ROS) elevate α -ketoglutarate (α KGM) levels, a metabolite in glutamine metabolism, which is processed by the enzyme nitrilase-like 2 ω -amidase (NIT2). They demonstrate that exposure to hydrogen peroxide (H_2O_2) oxidizes specific cysteine residues in NIT2, inhibiting its activity and leading to α KGM accumulation. Furthermore, the authors report that deleting NIT2 in endothelial cells of mice attenuates endothelial proliferation and angiogenesis while promoting cellular senescence. Their findings propose a role for NIT2 in endothelial glutamine metabolism and suggest a novel oxidative switch mechanism influencing vascular function.

Comment:

The manuscript is well-structured and presents solid data supporting a NIT2-dependent pathway that modulates α -ketoglutarate (α KG) levels in endothelial cells. The findings are logically presented, and the discussion effectively contextualizes their significance. However, several key questions remain regarding the physiological relevance of the proposed mechanism and its broader implications for vascular biology.

A primary concern is the analysis of NIT2's function in endothelial cells. The data do not convincingly establish whether the α KGM-NIT2- α KG pathway plays a meaningful role in this context. While the authors use different assays and models to show that NIT2 deficiency affects proliferation, senescence, and angiogenesis, the data quality and analytical resolution are insufficient for a thorough assessment of potential morphological abnormalities. This limitation is particularly evident in the mouse studies, where the morphological and cellular consequences of NIT2 deletion remain unclear.

Overall, the study would benefit from a more focused approach, with fewer but deeper analyses of the proposed metabolic pathway *in vivo*. On the other hand, the analysis of the single nucleotide variants does not add much to the central model of the work and could be left out.

Additional specific comments (major):

1. The physiological role of the NIT2 pathway in the vasculature is not convincingly established and the characterization of the mutant mice is unsatisfactory. This is important, as the reported anomalies appear rather mild. More details on the phenotype and its cellular nature are necessary to understand the relevance of the observations. As minimum, the authors should assess different stages of (retinal) vascular development and study different parameters such as endothelial cell number and proliferation to backup key claims.

2. The experiments in cultured endothelial cells suggest that NIT2 loss promotes senescence. Is senescence also observed in the endothelial NIT2 knockout mice?

3. The authors mention that α KGM rapidly degrades to the stable metabolite 2-hydroxy-5-oxo-proline in this context. One wonders what the function of this metabolite is in cells. Processes such as senescence take a relatively long time to develop, which raises the question whether 2-hydroxy-5-oxo-proline might be the biologically more relevant metabolite. Is it possible to stimulate (cultured) endothelial cells with 2-hydroxy-5-oxo-proline or α KGM to assess their direct effects?

4. Did the authors exclude that the α KGM-producing enzymes (KYATs) change in response to oxidative stress?

Referee #2:

Herrle et al. describe a redox-sensitive glutamine metabolism pathway in endothelial cells. They found that oxidative stress decreases α -ketoglutarate (α KG) while increasing its upstream metabolite α -ketoglutarate (α KGM), a reaction catalyzed by

the enzyme NIT2. Their study furthermore nicely shows that reduced NIT2 expression (due to a mutation) in humans correlates with higher α KGM levels and an increased risk of hypertension. Using metabolomic tracing experiments, they show that NIT2 functions in parallel with GLS1, the main aKG producing enzyme in ECs, in particular under conditions where GLS1 activity is low. Further, in vitro and in vivo experiments showed that NIT2 knockout impairs angiogenesis, depletes adenine levels, and reduces endothelial cell proliferation while promoting senescence. Finally, they demonstrated that inflammation and oxidative stress inactivate NIT2, with cysteine oxidation at Cys44 playing a key role.

This is a nice manuscript that describes the role of a highly understudied pathway (NIT2 mediated aKGM in angiogenesis/vascular biology). I have few questions:

1. The authors suggest that the mechanism through which NIT2 controls proliferation is adenine-dependent, yet they do not provide a potential explanation nor show any experiments using the obvious rescue approaches (supplementing aKG or altering the aKGM/aKG ratio). I propose to add those data. Moreover, the authors should discuss the potential mechanism through which NIT2 affects adenine.
2. The in vivo angiogenic imaging (and quantification thereof) is of very low quality. One cannot see any detail of the vasculature in the developing retina. Since these data are crucial to support the in vivo relevance, the images and quantification thereof should be improved. This should allow the reader to understand the phenotype better. Also, can the authors quantify EC proliferation in the developing retina?
3. Can the authors provide experimental evidence for the cellular localization of NIT2?
4. The authors use a pathological context (H₂O₂ or LPS) to alter NIT2 activity. Do they expect that NIT2 activity is also regulated under physiological conditions (such as the developing retina)?

Reply Letter to the editor and reviewers

Editor Comments:

Dear Dr. Klimmeck,

We thank you for your interest in our manuscript and for the opportunity to discuss it before for the revision. As guided by your advice, we carefully and extensively revised the manuscript according to the reviewers' comments.

In the following, please find the point by point reply to the reviewers. We have performed additional experiments and extended the manuscript to further strengthen our study. All changes are tracked in the manuscript text file.

Herrle et al: NIT2 and α -ketoglutarate - EMBOJ-2025-120471

Reviewer # 1

The manuscript is well-structured and presents solid data supporting a NIT2-dependent pathway that modulates α -ketoglutarate (α KG) levels in endothelial cells. The findings are logically presented, and the discussion effectively contextualizes their significance. However, several key questions remain regarding the physiological relevance of the proposed mechanism and its broader implications for vascular biology.

A primary concern is the analysis of NIT2's function in endothelial cells. The data do not convincingly establish whether the α KG-NIT2- α KG pathway plays a meaningful role in this context. While the authors use different assays and models to show that NIT2 deficiency affects proliferation, senescence, and angiogenesis, the data quality and analytical resolution are insufficient for a thorough assessment of potential morphological abnormalities. This limitation is particularly evident in the mouse studies, where the morphological and cellular consequences of NIT2 deletion remain unclear.

Overall, the study would benefit from a more focused approach, with fewer but deeper analyses of the proposed metabolic pathway *in vivo*.

On the other hand, the analysis of the single nucleotide variants does not add much to the central model of the work and could be left out.

Reply: We thank the reviewer for the time and effort spent in evaluating our manuscript. We performed additional experiments and have extended the analysis in the mouse studies, which are now included in the manuscript and the following replies.

Major comments:

1) The physiological role of the NIT2 pathway in the vasculature is not convincingly established and the characterization of the mutant mice is unsatisfactory. This is important, as the reported anomalies appear rather mild. More details on the phenotype and its cellular nature are necessary to understand the relevance of the observations. As minimum, the authors should assess different stages of (retinal) vascular development and study different parameters such as endothelial cell number and proliferation to backup key claims.

Reply: Following this suggestion, we have performed additional experiments to characterize the endothelial function of NIT2 *in vivo*. For the neonatal retina model we have focused on the endothelial cell-specific knockout model (CTR and *ecNit2*^{-/-} mice), where the earliest stage of retina development possible to analyse is P6 due to the injections with tamoxifen from P1-P3. [1] In fact, P5-P7 are the standard developmental stages studied in the field for evaluating sprouting angiogenesis. [1,2] To address endothelial cell proliferation in this angiogenic model, mice at P6 were injected with 5-ethynyl-2'-deoxyuridine (EdU) before being sacrificed and endothelial cells were labeled with isolectin B4. We provide in **Figure 1 reply letter** and **Figure 4 revised manuscript** images with higher resolution and larger magnification as compared to those in the original manuscript. Furthermore, we quantified EdU counts in vessels, total vessel length, vessel area and total number of junctions. All parameters were significantly decreased by the endothelial deletion of *Nit2*.

To address the role of endothelial NIT2 for angiogenesis *in vivo* in the adult stage, we analysed the choroidal neovascularization upon laser-induced injury. Endothelial cells in this type of angiogenesis penetrate through Bruch's membrane into the normally avascular subretinal space. [3] Briefly, adult mice received tamoxifen to delete endothelial *Nit2* and after the wash out time, four lesions were induced using an image-guided laser photocoagulation system. After 7 days, the mice were sacrificed and retinal pigment

epithelium/choroid/sclera was flat-mounted, stained with isolectin B4 and CD31, and imaged. Quantification of the area of the laser-induced lesions was performed using an established and constant threshold. Also in this model, vascularization was decreased upon deletion of Nit2 (**Figure 1L-M reply letter, Figure 4 revised manuscript**).

We include these data in Figure 4 of the revised manuscript and move the results from the aortic outgrowth assay and with the Nit2^{wt/wt} and Nit2^{ko/ko} mice into the supplementary material (**Suppl. Fig. 4 revised manuscript**).

2) The experiments in cultured endothelial cells suggest that NIT2 loss promotes senescence. Is senescence also observed in the endothelial NIT2 knockout mice?

Reply: This is a very interesting aspect that we approached experimentally by several means despite of the known challenges in mapping senescent cells in tissues such as the absence of specific markers and their relatively low abundance and vast heterogeneity. [4]

The standard staining for senescence is the β -Gal as shown for cells in figure 5F of the manuscript. We applied this technique in mouse retina and *en face* aortae of CTL and ecNit2^{-/-} mice. In both tissues, however, the signal was very weak and did not provide the resolution to assign it specifically to endothelial cells. As an alternative, we employed CellEventTM (Invitrogen, C10850), which is based on a fluorescent β -Gal substrate. Although the cellular resolution could be improved, the signal in aorta was weak and suffered from the auto-fluorescence of the aortic elastic fibers at the required excitation wavelength of 488nm (the dye is only available for this excitation). In mouse retina the assay gave some tissue staining, which was somewhat weak and not endothelial (**Fig. 2A reply letter**). As controls we also compared Nit2^{wt/wt} and Nit2^{ko/ko} young and aged mice (**Fig. 2B reply letter**) and included samples without CellEventTM. As the reviewer can appreciate, it was not possible to detect differences between young (3 months) nor aged (9 months) mice. We also looked at additional organs like heart, kidney and liver but could not detect any significant differences when comparing young versus aged mice. In particular, there was clear evidence in the liver and the kidney that the probe was unspecifically activated even in normal tissue.

To overcome the lack of direct senescence staining, we performed a MACE (massive analysis of cDNA ends) RNAseq experiment from endothelial cells specifically isolated from the carotid artery of CTL and ecNit2^{-/-} mice. MACE improves the sensitivity towards lowly expressed genes and requires only very little mRNA. Of the significantly differentially expressed genes, we filtered those annotated to senescence as recently published. [5,6] There was a significant increase in the expression of certain senescence-associated genes and SASP (senescence-associated secretory phenotype)-associated genes in ecNit2^{-/-} as compared to CTL mice. Among them, Galactosidase Beta 1 (Glb1, which often shows the largest changes in senescent data sets) [4] and Serpin2 (Serpin Family E Member 2, a SASP acting as serine protease inhibitor). However, there is no consensus senescence signature for murine endothelial cells *in vivo* and several genes in our analysis show low transcript numbers, despite of employing the MACE technique. Thus, we conclude that the staining techniques and gene signatures for senescence available do not provide sufficient resolution and sensitivity to make a statement concerning *in vivo* endothelial senescence in ecNit2^{-/-} mice.

3) The authors mention that α KGM rapidly degrades to the stable metabolite 2-hydroxy-5-oxo-proline in this context. One wonders what the function of this metabolite is in cells. Processes such as senescence take a relatively long time to develop, which raises the question whether 2-hydroxy-5-oxo-proline might be the biologically more relevant metabolite.

Is it possible to stimulate (cultured) endothelial cells with 2-hydroxy-5-oxo-proline or α KGM to assess their direct effects?

Reply: As requested, we determined the effect of 2-hydroxy-5-oxo-proline (300 μ M) added to cultured endothelial cells. As read-out we performed an RNAseq after 4 and 24 hours of exposure. The Principal component analysis (PCA) demonstrated that age of the culture, but not 2-hydroxy-5-oxo-proline did change gene expression and segregates PCA (**Fig. 3 reply letter**). This suggests that the compound is indeed inert and a metabolic end-product (like creatinine). These results are now included as **supplementary figure 5** in the manuscript.

4) Did the authors exclude that the α KGM-producing enzymes (KYATs) change in response to oxidative stress?

Reply: To answer this important question we have re-analysed our published metabolomics as well as the RNA expression data where HUVEC were exposed to 300 μ M H_2O_2 . KYATs are known for their function in tryptophan metabolism and none of the metabolites in this pathway were significantly altered by H_2O_2 as shown in **figure 4A-B reply letter**. Likewise, there were no changes in neither KYAT1 nor in KYAT3 expression (**Fig. 4C reply letter**) in response to H_2O_2 . Together, these results suggest that KYATs are not affected by oxidative stress. We add these results for the reviewer's benefit.

Reviewer #2

Herrle et al. describe a redox-sensitive glutamine metabolism pathway in endothelial cells. They found that oxidative stress decreases α -ketoglutarate (α KG) while increasing its upstream metabolite α -ketoglutarate (α KGM), a reaction catalyzed by the enzyme NIT2. Their study furthermore nicely shows that reduced NIT2 expression (due to a mutation) in humans correlates with higher α KGM levels and an increased risk of hypertension. Using metabolomic tracing experiments, they show that NIT2 functions in parallel with GLS1, the main α KG producing enzyme in ECs, in particular under conditions where GLS1 activity is low. Further, in vitro and in vivo experiments showed that NIT2 knockout impairs angiogenesis, depletes adenine levels, and reduces endothelial cell proliferation while promoting senescence. Finally, they demonstrated that inflammation and oxidative stress inactivate NIT2, with cysteine oxidation at Cys44 playing a key role.

This is a nice manuscript that describes the role of a highly understudied pathway (NIT2 mediated α KGM in angiogenesis/vascular biology). I have few questions:

Reply: We thank the reviewer for the time and effort spent in evaluating our manuscript. We performed additional experiments and have extended the analysis in the mouse studies, which are now included in the manuscript and the following replies.

1. The authors suggest that the mechanism through which NIT2 controls proliferation is adenine-dependent, yet they do not provide a potential explanation nor show any experiments using the obvious rescue approaches (supplementing α KG or altering the α KGM/ α KG ratio). I propose to add those data. Moreover, the authors should discuss the potential mechanism through which NIT2 affects adenine.

Reply: For the rescue approach we added cell-permeable α KG (dimethyl- α KG, dm- α KG [7]) at 100 μ M to NIT2^{-/-} or NIT2/GLS1^{-/-} cultured cells. We performed proliferation (**Fig. 5A reply letter**), senescence (**Fig. 5B reply letter**) and sprouting (**Fig. 5C reply letter**). In none of them, cell-permeable α KG rescued the phenotype of the knockout cells to a similar condition as NTC (non-targeted control). We add the data here for the reviewer's benefit.

Given that dm- α KG showed no rescue effect, we focused on adenine as an alternative metabolite that was decreased upon deletion of NIT2 in endothelial cells. To gain more insights into the metabolites altered by deletion of NIT2, we performed additional untargeted metabolomics using lung tissue and plasma from Nit2^{wt/wt} and Nit2^{ko/ko} mice (global, constitutive knockout to not be restricted to endothelial cells, **Table 1 and 2 reply letter**). Interestingly, there was a significant increase in several α -keto acids which are the suggested co-substrates for the KYAT enzymes in the transamination reaction that generates α KGM [8]. The most up-regulated metabolites in plasma of Nit2^{ko/ko} mice were indolelactate and 2-hydroxy-4-(methylthio)butanoic acid (KMBA), the α -keto acids of tryptophan and methionine, respectively (**Fig. 6 reply letter**).

KMBA is produced in the polyamine metabolism and part of the methionine salvage pathway. Strikingly, this pathway generates adenine from methyl-thio adenosine. The alterations in adenine and α -keto acids may provide the first hints that deletion of NIT2 uncouples the reactions interconnecting both pathways (**Fig. 7 reply letter**). Importantly, the specifics of the methionine salvage pathway have not yet been characterized in mammals. We, therefore, sense that the adenine mechanism is interesting but that an inclusion of these details would extent too far beyond the scope of the manuscript. We therefore add these results for the reviewer's benefit.

2. The in vivo angiogenic imaging (and quantification thereof) is of very low quality. One cannot see any detail of the vasculature in the developing retina. Since these data are crucial to support the in vivo relevance, the images and quantification thereof should be improved. This should allow the reader to understand the phenotype better. Also, can the authors quantify EC proliferation in the developing retina?

Reply: We have followed this request and performed additional experiments to improve the quality of the images and also to quantify EC proliferation. To address EC proliferation in the neonatal retina model, mice at P6 were injected with 5-ethynyl-2'-deoxyuridine (EdU) and endothelial cells were labeled with isolectin B4. We provide in **Figure 8 reply letter** and **Figure 4 revised manuscript** images with higher resolution and larger magnification as compared to those in the original manuscript. Furthermore, we quantified EdU counts in vessels, total vessel length, vessel area and total number of junctions. All parameters were significantly decreased by the endothelial deletion of Nit2.

We include these data in Figure 4 of the revised manuscript and move the results from the aortic outgrowth assay and with the Nit2^{wt/wt} and Nit2^{ko/ko} mice into the supplementary material (**Suppl. Fig. 4 revised manuscript**).

3. Can the authors provide experimental evidence for the cellular localization of NIT2?

Reply: To experimentally demonstrate the cellular localization of NIT2 we performed immunofluorescence for NIT2 in NTC and NIT2^{-/-} HUVEC where mitochondria were stained with MitoTracker Deep Red kit (Thermo Fisher, #M7512) and nuclei with DAPI. The staining showed a broad distribution across the cell, with low intensity, suggestive for cytosolic localization (**Fig. 9A reply letter**). Given the questionable quality of the images due to the antibody, we performed fractionation followed by Western blotting. Also this yielded NIT2 being located in the cytosol (positive for GAPDH). We further purified cytosolic and mitochondrial proteins from NTC HUVEC using a kit (Abcam #65320) that targets these two cellular compartments. Also with this technique, NIT2 showed a high localization in the cytosol with a minor fraction in the mitochondria (**Fig. 9C reply letter**). Next, we compared these experimental findings with prediction from BioGRID^{4.4} [9], which calculates scores for cellular localization based on the proteins interacting with NIT2. This tool yielded the highest scores for NIT2 for the cytosol followed by mitochondria (**Table 3 reply letter**). Therefore, we conclude that NIT2 is localized in the cytosol with a minor fraction in the mitochondria. We show these results for the reviewer's benefit.

4. The authors use a pathological context (H₂O₂ or LPS) to alter NIT2 activity. Do they expect that NIT2 activity is also regulated under physiological conditions (such as the developing retina)?

Reply: In our model, NIT2 activity was inhibited by acute oxidation with H₂O₂ as well as in activated granulocytes and in mice treated with LPS. These are conditions of strong oxidative stress. To substantiate this finding, we now have measured α KGM in supernatant and pellets of HUVEC exposed to granulocytes activated or not with zymosan (**Fig. 6K revised manuscript**). This experiment documents that not only NIT2 gets oxidized but also that this as a pathophysiological model results in the accumulation of α KGM (LC-MS/MS). We added this result to the revised manuscript (**Figure 10K reply letter, Fig. 6K revised manuscript**).

Concerning the physiological condition, we do not expect NIT2 activity to be inhibited. What might, however, very well be possible is that NIT2 is subjected to expression control under certain conditions. To investigate this, we studied NIT2 gene expression as determined by RT-qPCR in the developing mouse retina at different developmental states

(post-natal days 1, 3 and 6). In fact, NIT2 expression significantly increases at post-natal day 6, suggesting a role for NIT2 under physiological conditions (**Fig. 11A reply letter**). To address whether this increase in NIT2 expression is specific for endothelial cells, we FACS sorted CD144 positive cells (1:100, Biolegend, 138012, clone BV13) isolated from retinae from mice at P6 (proliferative EC) and P17 (quiescent EC). Interestingly, NIT2 expression (RT-qPCR) was increased at early stages, when cells are more proliferative (**Fig. 11B reply letter**). At later stages, when proliferation ceases, expression was lower. We show these results for the reviewer's benefit.

The figures for reviewers were removed.

References

- [1] M.E. Pitulescu, I. Schmidt, R. Benedito, R.H. Adams, Inducible gene targeting in the neonatal vasculature and analysis of retinal angiogenesis in mice, *Nat. Protoc.* 5 (2010) 1518–1534. <https://doi.org/10.1038/nprot.2010.113>.
- [2] G. Zarkada, J.P. Howard, X. Xiao, H. Park, M. Bizou, S. Leclerc, S.E. Künzel, B. Boisseau, J. Li, G. Cagnone, J.S. Joyal, G. Andelfinger, A. Eichmann, A. Dubrac, Specialized endothelial tip cells guide neuroretina vascularization and blood-retina-barrier formation, *Dev. Cell* 56 (2021) 2237–2251.e6. <https://doi.org/10.1016/j.devcel.2021.06.021>.
- [3] Y. Gong, J. Li, Y. Sun, Z. Fu, C.-H. Liu, L. Evans, K. Tian, N. Saba, T. Fredrick, P. Morss, J. Chen, L.E.H. Smith, Optimization of an Image-Guided Laser-Induced Choroidal Neovascularization Model in Mice, *PLoS One* 10 (2015) e0132643. <https://doi.org/10.1371/journal.pone.0132643>.
- [4] A.U. Gurkar, A.A. Gerencser, A.L. Mora, A.C. Nelson, A.R. Zhang, A.B. Lagnado, A. Enniful, C. Benz, D. Furman, D. Beaulieu, D. Jurk, E.L. Thompson, F. Wu, F. Rodriguez, G. Barthel, H. Chen, H. Phatnani, I. Heckenbach, J.H. Chuang, J. Horrell, J. Petrescu, J.K. Alder, J.H. Lee, L.J. Niedernhofer, M. Kumar, M. Königshoff, M. Bueno, M. Sokka, M. Scheibye-Knudsen, N. Neretti, O. Eickelberg, P.D. Adams, Q. Hu, Q. Zhu, R.A. Porritt, R. Dong, S. Peters, S. Victorelli, T. Pengo, T. Khaliullin, V. Suryadevara, X. Fu, Z. Bar-Joseph, Z. Ji, J.F. Passos, Spatial mapping of cellular senescence: emerging challenges and opportunities, *Nat. Aging* 3 (2023) 776–790. <https://doi.org/10.1038/s43587-023-00446-6>.
- [5] J. Wang, X. Zhou, P. Yu, J. Yao, P. Guo, Q. Xu, Y. Zhao, G. Wang, Q. Li, X. Zhu, G. Wei, W. Wang, T. Ni, A transcriptome-based human universal senescence index (hUSI) robustly predicts cellular senescence under various conditions, *Nat. Aging* 5 (2025) 1159–1175. <https://doi.org/10.1038/s43587-025-00886-2>.
- [6] D. Saul, R.L. Kosinsky, E.J. Atkinson, M.L. Doolittle, X. Zhang, N.K. LeBrasseur, R.J. Pignolo, P.D. Robbins, L.J. Niedernhofer, Y. Ikeno, D. Jurk, J.F. Passos, L.J. Hickson, A. Xue, D.G. Monroe, T. Tchkonja, J.L. Kirkland, J.N. Farr, S. Khosla, A new gene set identifies senescent cells and predicts senescence-associated pathways across tissues, *Nat. Commun.* 13 (2022) 4827. <https://doi.org/10.1038/s41467-022-32552-1>.
- [7] S.J. Parker, J. Encarnación-Rosado, K.E.R. Hollinshead, D.M. Hollinshead, L.J. Ash, J.A.K. Rossi, E.Y. Lin, A.S.W. Sohn, M.R. Philips, D.R. Jones, A.C. Kimmelman, Spontaneous hydrolysis and spurious metabolic properties of α -ketoglutarate esters, *Nat. Commun.* 12 (2021) 4905. <https://doi.org/10.1038/s41467-021-25228-9>.
- [8] J.T. Pinto, B.F. Krasnikov, S. Alcutt, M.E. Jones, T. Dorai, M.T. Villar, A. Artigues, J. Li, A.J.L. Cooper, Kynurenine aminotransferase III and glutamine transaminase L are identical enzymes that have cysteine S-conjugate β -lyase activity and can transaminate L-selenomethionine, *J. Biol. Chem.* 289 (2014) 30950–30961. <https://doi.org/10.1074/jbc.M114.591461>.
- [9] R. Oughtred, J. Rust, C. Chang, B.-J. Breitkreutz, C. Stark, A. Willems, L. Boucher, G. Leung, N. Kolas, F. Zhang, S. Dolma, J. Coulombe-Huntington, A. Chatr-Aryamontri, K. Dolinski, M. Tyers, The BioGRID database: A comprehensive biomedical resource of curated protein, genetic, and chemical interactions, *Protein Sci.* 30 (2021) 187–200. <https://doi.org/10.1002/pro.3978>.

Dear Dr Rezende,

Thank you for submitting your revised manuscript (EMBOJ-2025-120471R) to The EMBO Journal, as well for your patience with our feedback. Your amended study was sent back to the referees for their scientific reassessment, and we have received reports from both, which I enclose below. As you will see, the reviewers state that the work has been substantially enhanced by the revisions and they are now in favour of publication, pending minor amendments.

Thus, we are pleased to inform you that your manuscript has been accepted in principle for publication in The EMBO Journal.

Please carefully consider the remaining minor point raised by referee #2 by relativising claims related to the in vivo senescence phenotype results where appropriate.

Also, we now need you to take care of a number of issues related to formatting and data presentation as detailed below, which should be addressed at re-submission.

Please contact me at any time if you have additional questions related to below points.

As you might have seen on our web page, every paper at the EMBO Journal now includes a 'Synopsis', displayed on the html and freely accessible to all readers. The synopsis includes a 'model' figure as well as 2-5 one-short-sentence bullet points that summarize the article. I would appreciate if you could provide this figure and the bullet points.

Thank you for giving us the chance to consider your manuscript for The EMBO Journal. I look forward to your final revision.

Again, please contact me at any time if you need any help or have further questions.

Best regards,

Daniel Klimmeck

>> Please limit the keywords for your study to maximally five.

>> Author Contributions: Remove the author contributions information from the manuscript text. Note that CRediT has replaced the traditional author contributions section as of now because it offers a systematic machine-readable author contributions format that allows for more effective research assessment. and use the free text boxes beneath each contributing author's name to add specific details on the author's contribution.

More information is available in our guide to authors.
<https://www.embopress.org/page/journal/14602075/authorguide>

>> Adjust the title of the 'Disclosures' section to 'Disclosure and Competing Interests Statement'.

>> Rename the 'Summary' to 'Abstract'.

>> Remove the separate 'Limitations of this study' section, and integrate into the Discussion part.

>> Correct the order of the manuscript sections as follows: Abstract / Keywords / Introduction / Results / Discussion / Methods / Data Availability / Acknowledgements / Disclosure and Competing Interests Statement / References / Main Figure Legends / Tables / Expanded View Figure Legends.

>> Figure callouts: Please ensure that the six EV datasets are called out in sequential order in the main text.

>> Figures in separate files: all figures should be uploaded as individual, high-resolution figure files; main figures first, followed by EV figures.

>> Funding: please merge with Acknowledgements; the University of Frankfurt be added to the list of funders.

>> References: adjust the reference format to EMBO Journal format, 10 authors et al, and place References after the Disclosure and competing interests statement, before figure legends. Remove dois.

>> Reagents and Tools table: Please upload as a separate file using the existing template in the Guide For Authors, listing key reagents, experimental models, software and relevant equipment.

>> Data availability section: please remove the referee token for the GEO and PRIDE datasets and make sure that the data are made publicly accessible. Add hyperlinks to the datasets.

>> Author checklist: amend the Data Availability section by indicating GEO and PRIDE datasets and their section in the main text.

>> Please provide a completed source data checklist for the study as to the separate request e-mail.

>> Dataset EV legends: Please rename Tables 1 - 6 ("Dataset EV1" - Dataset EV6) and add a legend with the title and short description to each file in a separate tab or worksheet.

>> Consider additional changes and comments from our production team as indicated below:

- DAS:

- Figure legends:

1. Please define the annotated p values ****/****/**/* as well as provide the exact p-values for the same in the legend of figure 6B, J, L as appropriate.
2. Please note that the exact p values are not provided in the legends of figures 2D, F; 3C, D, F, G; 4B, H, I, J, K, M; 5B, C, G, H, J; 6H, K
3. Please indicate the statistical test used for data analysis in the legends of figures 2A, 3F, G, H; 4C, D; 6B, J, L.
4. Please note that information related to n is missing in the legends of figures 2D, F; 3C, D, F, G; 3H, 4B, 4C, D; 5B, C, D, E, G, H, J; 6B, H, J, K, L
5. Please note that the error bars are not defined in the legends of figures 2D, F; 3C, D, F, G, 4B, 5B, C, D, E, G, H, J; 6B, H, J, K, L

Please use the link below to submit your revision:

Referee #1:

The authors have submitted a revised version of their manuscript, in which they describe an NIT2-dependent pathway that modulates α -ketoglutarate (α KG) levels in endothelial cells. Although the physiological relevance of the mouse data remains somewhat unclear, I consider the manuscript improved and, overall, acceptable for publication.

Referee #2:

The authors have addressed my comments.

My only comment is regarding the lack of integration of the data into the revised manuscript. I propose that they integrate the data into the manuscript. Also, for it is not really clear why they leave the senescence data (reviewer#1) in the manuscript when they write that the in vivo evidence for senescence is not there. Leaving the in vitro data in without referring to the in vivo data does not make sense at all. Either the authors show the data or they take out everything?

The authors addressed the remaining editorial issues.

Dear Dr Rezende,

Thank you for submitting the revised version of your manuscript. I have now evaluated your amended manuscript and concluded that the remaining minor concerns have been sufficiently addressed.

I am thus pleased to inform you that your manuscript has been accepted for publication in the EMBO Journal.

Finally, we have noted that the submitted version of your article is also posted on the preprint platform bioRxiv. We would appreciate if you could alert bioRxiv on the acceptance of this manuscript at The EMBO Journal in order to allow for an update of the entry status. Thank you in advance!

Best regards,

Daniel Klimmeck

Daniel Klimmeck, PhD
Senior Editor
The EMBO Journal
EMBO
Postfach 1022-40
Meyerohofstrasse 1
D-69117 Heidelberg
contact@embojournal.org
